# Max-Min Off-Policy Actor-Critic Method Focusing on Worst-Case Robustness to Model Misspecification

**Takumi Tanabe**
University of Tsukuba & RIKEN AIP
Tsukuba, Ibaraki 305-8573, Japan
tanabe@bbo.cs.tsukuba.ac.jp

**Rei Sato**
University of Tsukuba & RIKEN AIP
Tsukuba, Ibaraki 305-8573, Japan
reisato@bbo.cs.tsukuba.ac.jp

**Kazuto Fukuchi**
University of Tsukuba & RIKEN AIP
Tsukuba, Ibaraki 305-8573, Japan
fukuchi@cs.tsukuba.ac.jp

**Jun Sakuma**
University of Tsukuba & RIKEN AIP
Tsukuba, Ibaraki 305-8573, Japan
jun@cs.tsukuba.ac.jp

**Youhei Akimoto**
University of Tsukuba & RIKEN AIP
Tsukuba, Ibaraki 305-8573, Japan
akimoto@cs.tsukuba.ac.jp

## Abstract

In the field of reinforcement learning, because of the high cost and risk of policy training in the real world, policies are trained in a simulation environment and transffered to the corresponding real-world environment. However, differences in the environments lead to model misspecification. Multiple studies report significant deterioration of policy performance in a real-world environment. In this study, we focus on scenarios involving a simulation environment with uncertainty parameters and the set of their possible values, called the uncertainty parameter set. The aim is to optimize the worst-case performance on the uncertainty parameter set to guarantee the performance in the corresponding real-world environment. To obtain a policy for the optimization, we propose an off-policy actor-critic approach called the Max-Min Twin Delayed Deep Deterministic Policy Gradient algorithm (M2TD3), which solves a max-min optimization problem using a simultaneous gradient ascent descent approach. Experiments in multi-joint dynamics with contact (MuJoCo) environments show that the proposed method exhibited a worst-case performance superior to several baseline approaches. Our implementation is publicly available (https://github.com/akimotolab/M2TD3).

## 1 Introduction

Applications of deep reinforcement learning (DRL) to control tasks that involving interaction with the real world are limited because of the need for interaction between the agent and the real-world environment [37]. This difficulty arises because of high interaction time, agent maintenance costs, and safety issues during the interaction. Without enough interactions, DRL tends to overfit to specific interaction histories and yields policies with poor generalizability and safety issues [29].

To solve this problem, simulation environments that estimate the characteristics of real-world environments are often employed, which enable enough interactions for effective DRL application. The policy is trained in the simulation environment and then transferred to the real-world environment [45].

36th Conference on Neural Information Processing Systems (NeurIPS 2022).

Moreover, because fast and general-purpose simulators, such as Open Dynamics Engine [34], Bullet [3] and multi-joint dynamics with contact (MuJoCo [38]), are available in the field of robotics, the cost of developing a simulation environment can sometimes be significantly lower than that for real-world interactions.

However, because a simulation environment is an estimation of the real-world environment, discrepancies exist between the two [2]. Even if they are entirely similar, discrepancies may still occur, e.g., because of robot degradation over time or weight changes because of part replacements [22]. Studies report significant degradation of policy trained in a simulation environment when transferred to the corresponding real-world environment [8]. The above discrepancies limit the use of simulator environments in practice and, thus, limit the use of DRL.

In this study, to guarantee performance in the corresponding real-world environment, we aim to obtain a policy that maximizes the expected reward under the worst-case scenario in the uncertainty parameter set $\Omega$. We focus on a scenario where (1) a simulator $\mathcal{M}_\omega$ is available, (2) the model uncertainty is parameterized by $\omega \in \Omega$, and $\omega$ is configurable in the simulation, and (3) the real-world environment is identified with some $\omega^* \in \Omega$, where $\omega^*$ is fixed during an episode; however, (4) $\omega^*$ is unknown, is not uniquely determined, or changes from episode to episode[1]. The model of uncertainty described in hypotheses (3) and (4) is referred to as the stationary uncertainty model, in contrast to the time-varying uncertainty model where $\omega^*$ can vary during an episode [28]. We develop an off-policy deep deterministic actor-critic approach to optimize the worst-case performance, called the Max-Min Twin Delayed Deep Deterministic Policy Gradient Algorithm (M2TD3). By extending the bi-level optimization formulation of the standard actor-critic method [30], we formulate our problem as a tri-level optimization, where the critic network models a value for each tuple of state, action, and uncertainty parameter, and the actor network models the policy of an agent and an estimate of the worst uncertainty parameter value. Based on the existing deep deterministic actor-critic approach [19; 7], we design an algorithm to solve the tri-level optimization problem by implementing several technical components to stabilize the training, including multiple uncertainty parameters, their refreshing strategies, the uncertainty parameter sampler for interaction, and the soft-min policy update. Numerical experiments on 19 MuJoCo tasks reveal the competitive and superior worst-case performance of our proposed approaches compared to those of several baseline approaches.

## 2   Preliminaries

We consider an episodic Markov Decision Process (MDP) [6] family $\mathcal{M}_\Omega = \{\mathcal{M}_\omega\}_{\omega \in \Omega}$, where $\mathcal{M}_\omega = \langle S, A, p_\omega, p_\omega^0, r_\omega, \gamma \rangle$ is the MDP with uncertainty parameter $\omega \in \Omega$. The state space $S$ and the action space $A$ are subsets of real-valued vector spaces. The transition probability density $p_\omega : S \times A \times S \to \mathbb{R}$, the initial state probability density $p_\omega^0 : S \to \mathbb{R}$, and the immediate reward $r_\omega : S \times A \to \mathbb{R}$ depend on $\omega$. The discount factor is denoted by $\gamma \in (0, 1)$.

We focus on an episodic task on $\mathcal{M}_\Omega$. Let $\mu_\theta : S \to A$ be a deterministic policy parameterized by $\theta \in \Theta$. Given an uncertainty parameter $\omega \in \Omega$, the initial state follows $s_0 \sim p_\omega^0$. At each time step $t \geqslant 0$, the agent observes state $s_t$, select action $a_t = \mu_\theta(s_t)$, interacts with the environment, and observes the next state $s_t' \sim p_\omega(\cdot \mid s_t, a_t)$, the immediate reward $r_t = r_\omega(s_t, a_t)$, and the termination flag $h_t$. Here, $h_t = 1$ if $s_t'$ is a terminal state or a predefined maximal time step for each past episode; otherwise, $h_t = 0$. If $h_t = 0$, let $s_{t+1} = s_t'$; otherwise, the next state is reset by $s_{t+1} \sim p_\omega^0$. For simplicity, we let $q_\omega(s_{t+1} \mid s_t, a_t)$ be the probability density of $s_{t+1}$ given $s_t$ and $a_t$. The discount reward of the trajectory starting from time step $t$ is $R_t = \sum_{k \geqslant 0} \gamma^k r_{t+k}$.

The action value function $Q^{\mu_\theta}(s, a, \omega)$ under $\omega$ is the expectation of $R_t$ starting with $s_t = s$ and $a_t = a$ under $\omega$; that is, $Q^{\mu_\theta}(s, a, \omega) = \mathbb{E}[R_t \mid s_t = s, a_t = a, s_{t+k+1} \sim q_\omega(\cdot \mid s_{t+k}, \mu_\theta(s_{t+k})) \; \forall k \geqslant 0]$, where the expectation is taken over $s_{t+k+1}$ for $k \geqslant 0$. Note that we introduce $\omega$ to the argument to explain the $Q$-value dependence on $\omega$; however, this is essentially the same definition as that for the

---

[1]Here are two examples where hypotheses (3) and (4) are reasonable. (I) We train a common controller of mass-produced robot manipulators that are slightly different due to errors in the manufacturing process ($\omega^*$ is fixed for each product, but varies in $\Omega$ from product to product). (II) We train a controller of a robot manipulator whose dynamics change over time because of aging, but each episode is short enough that the change in dynamics during an episode is negligible.

standard action value function $Q^{\mu_\theta}(s, a)$. The recursive formula is as follows:

$$Q^{\mu_\theta}(s, a, \omega) = r_\omega(s, a) + \gamma \mathbb{E}[Q^{\mu_\theta}(s', \mu_\theta(s'), \omega) \mid s' \sim q_\omega(\cdot \mid s, a)] \ . \tag{1}$$

We consider an off-policy reinforcement learning setting, where the agent interacts with $\mathcal{M}_\omega$ using a stochastic behavior policy $\beta : S \times A \to \mathbb{R}$. However, the target policy to be optimized, denoted by $\mu_\theta$, is deterministic. Under a fixed $\omega \in \Omega$, the standard objective of off-policy learning is to maximize the expected action value function over the stationary distribution $\rho^\beta$, such that

$$J_\omega(\theta) = \int_{s \in S} \rho^\beta(s \mid \omega) Q^{\mu_\theta}(s, \mu_\theta(s), \omega) \mathrm{d}s \ . \tag{2}$$

Here, $\rho^\beta(s \mid \omega) = \lim_{T \to \infty} \frac{1}{T} \sum_{t=0}^{T-1} \int_{s_0} q_{\omega,\beta}^t(s \mid s_0) p_\omega^0(s_0) \mathrm{d}s_0$ denotes the stationary distribution under $\beta$ and a fixed $\omega \in \Omega$, where the step-$t$ transition probability density $q_{\omega,\beta}^t$ is defined as $q_{\omega,\beta}^1(s' \mid s) = \int_{a \in A} q_\omega(s' \mid s, a) \beta(a \mid s) \mathrm{d}a$ and $q_{\omega,\beta}^t(s' \mid s) = \int_{\bar{s} \in S} q_{\omega,\beta}^{t-1}(\bar{s} \mid s) q_{\omega,\beta}^1(s' \mid \bar{s}) \mathrm{d}\bar{s}$.

We assume that the agent can interact with $\mathcal{M}_\omega$ for any $\omega \in \Omega$ during the training and can change $\omega$ after every episode, that is, when $h_t = 1$. Our objective is to obtain the $\mu_\theta$ that maximizes $J_\omega(\theta)$ under the worst environment $\omega \in \Omega$. Hence, we tackle the max-min optimization problem requiring $\max_{\theta \in \Theta} \min_{\omega \in \Omega} J_\omega(\theta)$.

## 3 M2TD3

We propose an off-policy actor-critic approach to obtain a policy that maximizes the worst-case performance on $\mathcal{M}_\Omega$, called M2TD3. Our approach is based on TD3, an off-policy actor-critic algorithm for a deterministic target policy. In TD3, the critic network $Q_\phi$ parameterized by $\phi \in \Phi$ is trained to approximate the $Q^{\mu_\theta}$ of $\mu_\theta$, whereas the actor network models $\mu_\theta$ with parameter $\theta \in \Theta$ and is trained to maximize $J_\omega(\theta)$. Moreover, $\omega$ is not considered and is fixed during the training. In this study, to obtain the $\mu_\theta$ that maximizes the performance under the worst $\omega$, we extend TD3 by formulating the objective as a maximin problem and introducing a simultaneous gradient ascent descent approach.

The main difficulty in the worst-case performance maximization is in estimating the worst-case performance. Because the objective is expected to be non-concave with respect to (w.r.t.) the uncertainty parameter because of deep actor-critic networks, solving this problem is considered intractable in general [4]. To stabilize the worst-case performance estimation, we introduce various techniques: multiple uncertainty parameters, an uncertainty parameter refresh strategy, and an uncertainty parameter sampler.

**Formulation** We can formulate our objective as the following tri-level optimization:

$$\max_{\theta \in \Theta} \min_{\hat{\omega} \in \Omega} J(\theta, \hat{\omega}; \phi^*) \quad \text{s.t.} \quad \phi^* \in \operatorname*{argmin}_{\phi \in \Phi} L(\phi; \theta) \ , \tag{3}$$

where $J$ and $L$ are defined below. Note that this tri-level optimization formulation is an extension of the bi-level optimization formulation of the actor-critic approach proposed in [30].

We introduce a probability density $\alpha : \Omega \to \mathbb{R}$, from which an $\omega$ to be used in the interaction during the training phase is drawn for each episode. The relationship between $\hat{\omega}$ and $\alpha$ is similar to that between $\theta$ and $\beta$: namely, $\hat{\omega}$ and $\theta$ are the parameters to be optimized while $\alpha$ and $\beta$ are introduced for exploration. Let $\rho_\alpha^\beta(s) = \int_{\omega \in \Omega} \rho^\beta(s \mid \omega) \alpha(\omega) \mathrm{d}\omega$ and $\rho_\alpha^\beta(s, a, \omega) = \beta(a \mid s) \rho^\beta(s \mid \omega) \alpha(\omega)$ be the stationary distribution of $s$ and the joint stationary distribution of $(s, a, \omega)$, respectively, under $\alpha$ and $\beta$.

The critic loss function $L(\phi; \theta)$ is designed to simulate the Q-learning algorithm. Let $T_{\mu_\theta}$ be the function satisfying $T_{\mu_\theta}[Q](s, a, \omega) = r_\omega(s, a) + \gamma \int_{s' \in S} Q(s', \mu_\theta(s'), \omega) q_\omega(s' \mid s, a) \mathrm{d}s'$. Then, (1) states that $Q^{\mu_\theta}$ is the solution to $Q = T_{\mu_\theta}[Q]$. Therefore, the critic is trained to minimize the difference between $Q_\phi$ and $T_{\mu_\theta}[Q_\phi]$. This is achieved by minimizing

$$L(\phi; \theta) := \int_{s \in S} \int_{a \in A} \int_{\omega \in \Omega} (T_{\mu_\theta}[Q_\phi](s, a, \omega) - Q_\phi(s, a, \omega))^2 \rho_\alpha^\beta(s, a, \omega) \mathrm{d}s \mathrm{d}a \mathrm{d}\omega \ . \tag{4}$$

The max-min objective function of the actor network

$$J(\theta, \hat{\omega}; \phi) := \int_{s \in S} Q_\phi(s, \mu_\theta(s), \hat{\omega}) \rho_\alpha^\beta(s) \mathrm{d}s \tag{5}$$

measures the performance of $\mu_\theta$ under the uncertainty parameter $\hat{\omega} \in \Omega$ approximated using the critic network $Q_\phi$ instead of the ground truth $Q^{\mu_\theta}$. Similar to the standard actor-critic approach, we expect $Q_\phi$ to approach $Q^{\mu_\theta}$ as the critic loss is minimized. Therefore, we expect $J(\theta, \hat{\omega}; \phi^*)$ approximates $\int_{s \in S} Q^{\mu_\theta}(s, \mu_\theta(s), \hat{\omega}) \rho_\alpha^\beta(s) \mathrm{d}s$ once we obtain $\phi^* \approx \operatorname{argmin}_{\phi \in \Phi} L(\phi; \theta)$.

Particularly, even if $Q_{\phi^*} = Q^{\mu_\theta}$, our objective function $J(\theta, \hat{\omega}; \phi^*)$ differs from $J_\omega(\theta)$ with $\omega = \hat{\omega}$ in (2), in that the stationary distribution $\rho^\beta(s \mid \omega)$ under fixed $\omega$ in $J_\omega$ is replaced with $\rho_\alpha^\beta(s)$ under $\omega \sim \alpha$. However, this change allows us to effectively utilize the replay buffer, which stores the interaction history, to approximate the objective $J(\theta, \hat{\omega}; \phi)$. Moreover, if $\alpha$ is concentrated at $\hat{\omega}$, $J(\theta, \hat{\omega}; \phi^*)$ coincides with $J_{\omega=\hat{\omega}}(\theta)$.

**Algorithmic Framework**    The framework of M2TD3 is designed to solve the tri-level optimization (3). The overall framework of M2TD3 follows that of TD3. In each episode, an uncertainty parameter $\omega$ is sampled from $\alpha$. The training agent interacts with the environment $\mathcal{M}_\omega$ using behavior policy $\beta$. At each time step $t$, the transition $(s_t, a_t, r_t, s_t', h_t, \omega)$ is stored in the replay buffer, which is denoted by $B$. The critic network $Q_\phi$ is trained at every time step, to minimize (4), with a mini-batch being drawn uniformly randomly from $B$. The actor network $\mu_\theta$ and the worst-case uncertainty parameter $\hat{\omega}$ are trained in every $T_{\text{freq}}$ step to optimize (5). Note that a uniform random sample $(s, a, \omega)$ taken from $B$ can be regarded as a sample from the stationary distribution $\rho_\alpha^\beta(s, a, \omega)$. Similarly, $s$ taken from $B$ is regarded as a sample from $\rho_\alpha^\beta(s)$. These facts allow approximation of the expectations in (4) and (5) using the Monte Carlo method with mini-batch samples uniformly randomly taken from $B$. An algorithmic overview of the proposed method is summarized in Algorithm 1. We have underlined the differences from the general off-policy actor-critic method in the episodic settings. A detailed description of M2TD3 is summarized in Algorithm 2 in Appendix A. As shown from Algorithm 1, the differences between the proposed method and the general off-policy actor-critic are as follows: (1) the introduction of an uncertainty parameter sampler, (2) definition and updating method of critic, and (3) updating method of actor and uncertainty parameters.

---

**Algorithm 1** Algorithmic overview of the proposed method

---

 1: Draw uncertainty parameter $\omega \sim \alpha_0$
 2: Observe initial state $s \sim p_\omega^0$
 3: **for** t = 1 to $T_{\text{max}}$ **do**
 4:    # interaction
 5:    Select action $a \sim \beta_t(s)$
 6:    Interact with $\mathcal{M}_\omega$ with $a$, observe next state $s'$, immediate reward $r$ and termination flag $h$
 7:    Store transition tuple $(s, a, r, s', h, \omega)$ in $B$
 8:    **if** $h = 1$ **then**
 9:       Reset uncertainty parameter $\omega \sim \alpha_t$
10:       Observe initial state $s \sim p_\omega^0$
11:    **else**
12:       Update current state $s \leftarrow s'$
13:    **end if**
14:    # learning
15:    Sample mini-batch $\{(s_i, a_i, r_i, s_i', h_i, \omega_i)\}$ of M transitions uniform-randomly from $B$
16:    Update the critic network by optimizing Equation (6)
17:    Update the actor network and the uncertainty parameter by optimizing Equation (7)
18:    Update uncertainty parameter sampler $\alpha$
19: **end for**

---

**Critic Update**    The critic network update, namely, minimization of (4), is performed by implementing the TD error-based approach. The concept is as follows. Let $\{(s_i, a_i, r_i, s_i', \omega_i)\}_{i=1}^M \subset B$ be the mini-batch, and let $\theta_t$ and $\phi_t$ be the actor and critic parameters at time step $t$, respectively. For each tuple, $T_{\mu_{\theta_t}}[Q_{\phi_t}](s_i, a_i, \omega_i)$ is approximated by $y_i = r_i + \gamma \cdot Q_{\phi_t}(s_i', \mu_{\theta_t}(s_i'), \omega_i)$. Then, $L(\phi_t; \theta_t)$

in (4) is approximated by the mean square error $\tilde{L}(\phi_t)$, such that

$$\tilde{L}(\phi_t) = \frac{1}{M} \sum_{i=1}^{M} (y_i - Q_{\phi_t}(s_i, a_i, \omega_i))^2 \ . \tag{6}$$

Then, the update follows $\phi_{t+1} = \phi_t - \lambda_\phi \nabla_\phi \tilde{L}(\phi_t)$.

We incorporate the techniques introduced in TD3 to stabilize the training: clipped double Q-learning, target policy smoothing regularization, and target networks. Additionally, we introduce the smoothing regularization for the uncertainty parameter so that the critic is smooth w.r.t. $\hat{\omega}$. Algorithm 3 in Appendix A summarizes the critic update and the details of each technique are available in [7].

**Actor Update**    Updating of the actor as well as the uncertainty parameter, namely, maximin optimization of (5), is performed via simultaneous gradient ascent descent [27], summarized in Algorithm 4 in Appendix A. Let $\phi_t$, $\theta_t$ and $\hat{\omega}_t$ be the critic, policy, and the uncertainty parameters, respectively, at time step $t$. Instead of the optimal $\phi^*$ in (3), we use $\phi_t$ as its estimate. With the mini-batch $\{s_i\}_{i=1}^{M} \subset B$, we approximate the max-min objective (5) follows:

$$\tilde{J}_t(\theta, \hat{\omega}) = \frac{1}{M} \sum_{i=1}^{M} Q_{\phi_t}(s_i, \mu_\theta(s), \hat{\omega}) \ . \tag{7}$$

We update $\theta$ and $\hat{\omega}$ as $\theta_{t+1} = \theta_t + \lambda_\theta \nabla_\theta \tilde{J}_t(\theta_t, \hat{\omega}_t)$ and $\hat{\omega}_{t+1} = \hat{\omega}_t - \lambda_\omega \nabla_{\hat{\omega}} \tilde{J}_t(\theta_t, \hat{\omega}_t)$, respectively.

For higher-stability performance, we introduce multiple uncertainty parameters. The motivation is twofold. One is to deal with multiple local minima of $J_t(\theta, \hat{\omega}; \phi^*)$ w.r.t. $\hat{\omega}$. As the critic network becomes generally non-convex, there may exist multiple local minima of $J_t(\theta, \hat{\omega}; \phi^*)$ w.r.t. $\hat{\omega}$. Once $\hat{\omega}$ is stacked at a local minimum point, e.g., $\hat{\omega}^*$, $\theta$ may be trained to be robust around a neighborhood of $\hat{\omega}^*$ and to perform poorly outside that neighborhood. The other is that the maximin solution $(\theta^*, \phi^*)$ of $J_t(\theta, \hat{\omega}; \phi^*)$ is not a saddle point, which occurs when the objective is non-concave in $\hat{\omega}$ [14]. Here, more than one $\hat{\omega}$ is necessary to approximate the worst-case performance $\min_{\hat{\omega}} J_t(\theta, \hat{\omega}; \phi^*)$ around $(\theta^*, \phi^*)$ and a standard simultaneous gradient ascent descent method fails to converge. To relax this defect of simultaneous gradient ascent descent methods, we maintain multiple candidates for the worst $\hat{\omega}$, denoted by $\hat{\omega}_1, \ldots, \hat{\omega}_N$. Therefore, replacing $\min_{\hat{\omega}} J(\theta, \hat{\omega}; \phi^*)$ with $\min_{k=1,\ldots,N} J(\theta, \hat{\omega}_k; \phi^*)$ in (5) has no effect on the optimal $\theta$; however, this change does affect the training behavior. Our update follows the simultaneous gradient ascent descent on $\min_{k=1,\ldots,N} J_t(\theta, \hat{\omega}_k)$: $\theta \leftarrow \theta + \lambda_\theta \nabla_\theta (\min_{k=1,\ldots,N} J_t(\theta, \hat{\omega}_k))$ and $\hat{\omega}_k \leftarrow \hat{\omega}_k - \lambda_\omega \nabla_{\hat{\omega}_k} (\min_{\ell=1,\ldots,N} J_t(\theta, \hat{\omega}_\ell))$ for $k = 1, \ldots, N$. Hence, $\theta$ is updated against the worst $\hat{\omega}_k$, and only the worst $\hat{\omega}_k$ is updated because the gradient w.r.t. the other $\hat{\omega}_k$ is zero.

For multiple uncertainty parameters to be effective, they must be distant from each other. Moreover, all are expected to be selected as the worst parameters with non-negligible frequencies. Otherwise, the advantage of having multiple uncertainty parameters is lessened. From this perspective, we introduce a refreshing strategy for uncertainty parameters. Namely, we resample $\hat{\omega}_k \sim \mathcal{U}(\Omega)$ if one of the following scenarios is observed: there exists $\hat{\omega}_\ell$ such that the distance $d_{\hat{\omega}}(\hat{\omega}_k, \hat{\omega}_\ell) \leqslant d_{\text{thre}}$; the frequency $p_k$ of $\hat{\omega}_k$ being selected as the worst-case during the actor update is no greater than $p_{\text{thre}}$.

**Uncertainty Parameter Sampler**    The uncertainty parameter sampler $\alpha$ controls the exploration-exploitation trade-off in the uncertainty parameter. Exploration in $\Omega$ is necessary to train the critic network. If the critic network is not well trained over $\Omega$, it is difficult to locate the worst $\hat{\omega}$ correctly. On the other hand, for $J(\theta, \hat{\omega}; \phi^*)$ in (5) to coincide with $J_\omega(\theta)$ for $\omega = \hat{\omega}$ in (2), we require $\alpha$ to be concentrated at $\hat{\omega}$. Otherwise, the optimal $\theta$ for $J(\theta, \hat{\omega}; \phi^*)$ may deviate from that of $J_\omega(\theta)$. We design $\alpha$ as follows. For the first $T_{\text{rand}}$ steps, we sample the uncertainty parameter $\omega$ uniformly randomly on $\Omega$, i.e., $\alpha = \mathcal{U}(\Omega)$. Then, we set $\alpha = \sum_{k=1}^{N} p_k \cdot \mathcal{N}(\hat{\omega}_k, \Sigma_\omega)$, where $\Sigma_\omega$ is the predefined covariance matrix. We decrease $\Sigma_\omega$ as the time step increases. The rationale behind the gradual decrease of $\Sigma_\omega$ is that the training of the critic network is still important after $T_{\text{rand}}$ to make the estimation of the worst-case uncertainty parameter accurate. Details are provided in Appendix C.

## 4    Related Work

Methods to handle model misspecification include (1) robust policy searching [26; 31; 23; 17; 12; 22], (2) transfer learning [36; 43; 32], (3) domain randomization (DR) [37], and (4) approaches minimizing

worst-case sub-optimality gap [5; 21]. Robust policy searching includes methods that aim to obtain the optimal policy under the worst possible disturbance or model misspecification, i.e., maximization of $\min_{\omega \in \Omega} J_\omega(\theta)$ for an explicitly or implicitly defined $\Omega$. Transfer learning is a method in which a policy is trained on source tasks and then fine-tuned through interactions while performing the target task. DR is a method in which a policy is trained on source tasks that are sampled randomly from $\mathcal{M}_\Omega$. There are two types of DR approaches: vanilla and guided. In vanilla DR [37], the policy is trained on source tasks that are randomly sampled from a pre-defined distribution on $\Omega$. In guided DR [44], the policy is trained on source tasks; however, the $\omega$ distribution is guided towards the target task. Because we do not assume access to the target task, transfer learning and many guided DR approaches are outside the scope of this work. Some guided DR approaches, such as active domain randomization [24], do not access the target task or consider worst-case optimization either. Vanilla DR can be applied to the setting considered here. However, the objective of vanilla DR differs from the present aim, i.e., to maximize the performance averaged over $\Omega$, namely, $\mathbb{E}_\omega[J_\omega(\theta) \mid \omega \sim \mathcal{U}(\Omega)]$, if the sampling distribution is $\mathcal{U}(\Omega)$. The approaches minimizing the worst-case sub-optimality gap [5; 21] do not optimize the worst-case performance, instead attempt to obtain a policy that generalizes well on the uncertainty parameter set while avoiding too conservative performance, which often attributes to the worst-case performance optimization.

Some robust policy search methods, some adopt an adversarial approach to policy optimization. For example, Robust Adversarial Reinforcement Learning (RARL) [31] models the disturbance caused by an external force produced by an adversarial agent, and alternatively trains the protagonist and adversarial agents. Robust Reinforcement Learning (RRL) [26] similarly models the disturbance but has not been applied to the DRL framework. Minimax Multi-Agent Deep Deterministic Policy Gradient (M3DDPG) [17] has been designed to obtain a robust policy for multi-agent settings. This method is applicable to the setting targeted in this study if only two agents are considered. The above approaches frame the problem as a zero-sum game between a protagonist agent attempting to optimize $\mu_\theta$ and an adversarial agent attempting to minimize the protagonist's performance, hindering the protagonist by generating the worst possible disturbance. Adv-Soft Actor Critic (Adv-SAC) [13] learns policies that are robust to both internal disturbances in the robot's joint space and those from other robots. Recurrent SAC (RSAC) [42] introduces POMDPs to treat the uncertainty parameter as an unobservable state. A DDPG based approach robustified by applying stochastic gradient langevin dynamics is proposed under the noisy robust MDP setting [15]. However, it has been reported in [42; 15] that their worst-case performances are sometimes even worse than their baseline non-robust approaches. State-Adversarial-DRL (SA-DRL) [48], and alternating training with learned adversaries (ATLA) [47] improve the robustness of DRL agents by using an adversary that perturbs the observations in SAMDP framework that characterizes the decision-making problem in an adversarial attack on state observations. They do not address the model misspecification in the reward and transition functions.

Other approaches attempt to estimate the robust value function, i.e., a value function under the worst uncertainty parameter. Among them, Robust Dynamic Programming (DP) [12] is a DP approach, while the Robust Options Policy Iteration (ROPI) [23] incorporates robustness into option learning [35], which allows agents to learn both hierarchical policies and their corresponding option sets. ROPI is a type of Robust MDP approach [39; 20]. Robust Maximum A-posteriori Policy Optimization (R-MPO) [22] incorporates robustness in MPO [1].

Neither Robust MDP nor R-MPO require interaction with $M_\omega$ for an arbitrary $\omega \in \Omega$. This can be advantageous in scenarios where the design of a simulator valid over $\Omega$ is tedious. However, these methods typically require additional assumptions for computing the worst-case for each value function update, such as the finiteness of the state-action space and/or the finiteness of $\Omega$. To apply R-MPO to our setting, finite scenarios from $\Omega$ for training are sampled, but the choice of the training scenarios affects the worst-case performance.

Additionally, to optimize the worst-case performance in the field of offline RL [40; 46; 41], offline RL attempts to obtain robust measures by introducing the principle of pessimism. Some studies that introduce the principle of pessimism in the model-free context [40; 41] and in the actor-critic context [46]. However, these do not compete with this study due to differences in motivation and because offline RL requires additional assumptions in the datasets [46; 41] and realizability [40].

Table 5 in Appendix D summarizes the robust policy search method taxonomy. Our proposed approach, M2TD3, considers the uncertainties in both the reward function and transition probability.

The state and action spaces, as well as the uncertainty set, are assumed to be continuous. Table 6 in Appendix D compares the related methods considering their action value function definitions. Methods that do not explicitly utilize the action value function are interpreted from the objective functions. Both M3DDPG and M2TD3 maintain the value function for the tuple of $(s, a, \omega)$. However, M3DDPG takes the worst $\omega'$ on the right-hand side, yielding a more conservative policy than M2TD3 for our setting because we assume a stationary uncertainty model, whereas M3DDPG assumes a time-varying uncertainty model in nature [28]. The value functions in RARL and M2TD3 are identical if $\omega$ is fixed. Particularly, $\omega$ is not introduced to the RARL value function because it does not simultaneously optimize $\mu_\theta$ and the estimate of the worst $\omega$. RARL repeats the $\theta$ and $\hat{\omega}$ optimizations alternatively. In contrast, M2TD3 optimizes $\theta$ and $\hat{\omega}$ simultaneously in a manner similar to the training processes of generative adversarial networks (GANs) [9]. Hence, we require an action value for each $\omega \in \Omega$.

A difference exists between the optimization strategies of RARL and M2TD3. As noted in the aforementioned section, both methods attempt to maximize the worst-case performance $\min_{\omega \in \Omega} J_\omega(\theta)$. Conceptually, RARL repeats $\theta \leftarrow \mathrm{argmax}_\theta J_\omega(\theta)$ and $\omega \leftarrow \mathrm{argmin}_\omega J_\omega(\theta)$. However, this optimization strategy fails to converge even if the objective function $(\theta, \omega) \mapsto J_\omega(\theta)$ is concave-convex. As an example, consider the function $(x, y) \mapsto y^2 - x^2 + \alpha xy$. The RARL optimization strategy reads $x \leftarrow (\alpha/2)y$ and $y \leftarrow -(\alpha/2)x$, which causes divergence if $\alpha > 2$[2]. Alternating updates are also employed in other approaches such as Adv-SAC, SA-DRL, and ATLA. These share the same potential issue as RARL. M2TD3 attempts to alleviate this divergence problem by applying the gradient-based max-min optimization method, which has been employed in GANs and other applications and analyzed for its convergence [25; 18].

## 5 Experiments

In this study, we conducted experiments on the optimal control problem using MuJoCo environments. Hence, we demonstrated the problems of existing methods and assessed the worst-case performance and average performance of the policy trained by M2TD3 for different continuous control tasks.

**Baseline Methods**    We summarize the baseline methods adapted to our experiment setting, namely, DR, RARL, and M3DDPG.

DR: The objective of DR is to maximize the expected cumulative reward for the distribution $\alpha = \mathcal{U}(\Omega)$ of $\omega$. In each training episode, $\omega$ is drawn randomly from $\mathcal{U}(\Omega)$, and the agent neglects $\omega$ when training and performing the standard DRL. For a fair comparison, we implemented DR with TD3 as the baseline DRL method.

RARL: We adapted RARL to our scenario by setting $\mu_{\hat{\omega}} : s \mapsto \hat{\omega}$ to the antagonist policy. RARL was regarded as optimizing (2) for the worst $\omega \in \Omega$; hence, the objective was the same as that of M2TD3. Particularly, the main technical difference between M2TD3 and RARL is in the optimization strategy as described in Section 4. The original RARL is implemented with Trust Region Policy Optimization (TRPO) [33], but for a fair comparison, we implemented it with DDPG, denoted as RARL (DDPG). The experimental results of RARL (TRPO), the original RARL, are provided in Appendix F.

M3DDPG: By considering a two-agent scenario and a state-independent policy $\mu_{\hat{\omega}} : s \mapsto \hat{\omega}$ as the opponent-agent's policy, we adapted M3DDPG to our setting. M3DDPG is different from M2TD3 (and M2-DDPG below) even under a state-independent policy as described in Section 4. Because of the difficulty in replacing DDPG with TD3 in the M3DDPG framework, we used DDPG as the baseline DRL.

Additionally, for comparison, we implemented our approach with DDPG instead of TD3, denoted by M2-DDPG. We also implemented a variant of M2TD3, denoted as SoftM2TD3, performing a "soft" worst-case optimization to achieve better average performance while considering the worst-case performance. The detail of SoftM2TD3 is described in Appendix B.

---

[2]A simple way to mitigate this issue would be to early-stop the optimization for each step of RARL. However, the problem remains. The protagonist agent in RARL does not consider the uncertainty parameter in the critic. This leads to a non-stationary training environment for both protagonist and adversarial agents. As the learning in a non-stationary environment is generally difficult [11], RARL will be unstable. The instability of RARL has also been investigated in the linear-quadratic system settings [49].

Table 1: Avg. $\pm$ std. error of worst-case performance $R_{\mathrm{worst}}(\mu)$ over 10 trials for each approach

| Environment | M2TD3 | SoftM2TD3 | M2-DDPG | M3DDPG | RARL (DDPG) | DR (TD3) |
|---|---|---|---|---|---|---|
| Ant 1 ($\times 10^3$) | $3.84 \pm 0.10$ | $\mathbf{4.08 \pm 0.15}$ | $1.28 \pm 0.19$ | $0.49 \pm 0.12$ | $-1.24 \pm 0.10$ | $3.51 \pm 0.08$ |
| Ant 2 ($\times 10^3$) | $\mathbf{4.13 \pm 0.11}$ | $3.92 \pm 0.14$ | $0.95 \pm 0.20$ | $-0.25 \pm 0.13$ | $-1.77 \pm 0.09$ | $1.64 \pm 0.13$ |
| Ant 3 ($\times 10^3$) | $\mathbf{0.10 \pm 0.10}$ | $0.07 \pm 0.20$ | $-1.13 \pm 0.28$ | $-1.38 \pm 0.22$ | $-2.38 \pm 0.07$ | $-0.32 \pm 0.03$ |
| HalfCheetah 1 ($\times 10^3$) | $3.14 \pm 0.10$ | $\mathbf{3.24 \pm 0.08}$ | $2.24 \pm 0.25$ | $-0.13 \pm 0.12$ | $-0.55 \pm 0.02$ | $3.19 \pm 0.08$ |
| HalfCheetah 2 ($\times 10^3$) | $2.61 \pm 0.16$ | $\mathbf{2.82 \pm 0.16}$ | $2.54 \pm 0.23$ | $-0.58 \pm 0.06$ | $-0.70 \pm 0.05$ | $2.12 \pm 0.13$ |
| HalfCheetah 3 ($\times 10^3$) | $0.93 \pm 0.21$ | $\mathbf{1.53 \pm 0.23}$ | $1.20 \pm 0.22$ | $-0.66 \pm 0.08$ | $-0.81 \pm 0.07$ | $1.09 \pm 0.06$ |
| Hopper 1 ($\times 10^2$) | $\mathbf{6.21 \pm 0.45}$ | $5.98 \pm 0.23$ | $5.38 \pm 0.43$ | $4.14 \pm 0.60$ | $3.32 \pm 0.78$ | $5.28 \pm 2.55$ |
| Hopper 2 ($\times 10^2$) | $5.33 \pm 0.28$ | $\mathbf{5.79 \pm 0.29}$ | $4.30 \pm 0.57$ | $2.58 \pm 0.29$ | $3.34 \pm 0.89$ | $4.68 \pm 0.15$ |
| Hopper 3 ($\times 10^2$) | $\mathbf{2.84 \pm 0.25}$ | $1.98 \pm 0.22$ | $2.25 \pm 0.29$ | $0.73 \pm 0.11$ | $1.64 \pm 0.46$ | $2.10 \pm 0.35$ |
| HumanoidStandup 1 ($\times 10^4$) | $9.33 \pm 0.70$ | $9.49 \pm 0.81$ | $8.09 \pm 0.92$ | $8.00 \pm 0.78$ | $5.29 \pm 0.45$ | $\mathbf{9.68 \pm 0.60}$ |
| HumanoidStandup 2 ($\times 10^4$) | $6.50 \pm 0.70$ | $\mathbf{7.94 \pm 0.90}$ | $6.24 \pm 0.54$ | $6.37 \pm 0.72$ | $5.78 \pm 0.73$ | $7.31 \pm 0.78$ |
| HumanoidStandup 3 ($\times 10^4$) | $\mathbf{6.20 \pm 0.64}$ | $5.99 \pm 0.37$ | $5.96 \pm 0.58$ | $6.01 \pm 0.38$ | $5.54 \pm 0.76$ | $5.41 \pm 0.34$ |
| InvertedPendulum 1 ($\times 10^2$) | $\mathbf{8.22 \pm 1.13}$ | $6.53 \pm 1.36$ | $6.49 \pm 1.33$ | $1.09 \pm 0.71$ | $1.53 \pm 0.64$ | $3.18 \pm 1.10$ |
| InvertedPendulum 2 ($\times 10^2$) | $\mathbf{3.56 \pm 1.32}$ | $1.36 \pm 0.30$ | $1.10 \pm 0.62$ | $0.02 \pm 0.00$ | $0.02 \pm 0.00$ | $0.57 \pm 0.02$ |
| Walker 1 ($\times 10^3$) | $2.83 \pm 0.39$ | $\mathbf{3.02 \pm 0.22}$ | $1.19 \pm 0.17$ | $0.89 \pm 0.18$ | $0.09 \pm 0.02$ | $2.19 \pm 0.40$ |
| Walker 2 ($\times 10^3$) | $\mathbf{3.14 \pm 0.39}$ | $2.64 \pm 0.43$ | $0.85 \pm 0.12$ | $0.39 \pm 0.11$ | $0.06 \pm 0.04$ | $2.31 \pm 0.50$ |
| Walker 3 ($\times 10^3$) | $1.94 \pm 0.40$ | $\mathbf{2.00 \pm 0.35}$ | $0.82 \pm 0.13$ | $0.28 \pm 0.09$ | $0.00 \pm 0.02$ | $1.32 \pm 0.34$ |
| Small HalfCheetah 1 ($\times 10^3$) | $5.27 \pm 0.12$ | $5.07 \pm 0.14$ | $4.51 \pm 0.18$ | $1.26 \pm 0.38$ | $-0.52 \pm 0.02$ | $\mathbf{6.76 \pm 0.18}$ |
| Small Hopper 1 ($\times 10^3$) | $2.88 \pm 0.32$ | $2.23 \pm 0.32$ | $1.40 \pm 0.19$ | $1.39 \pm 0.21$ | $0.51 \pm 0.12$ | $\mathbf{3.42 \pm 0.11}$ |

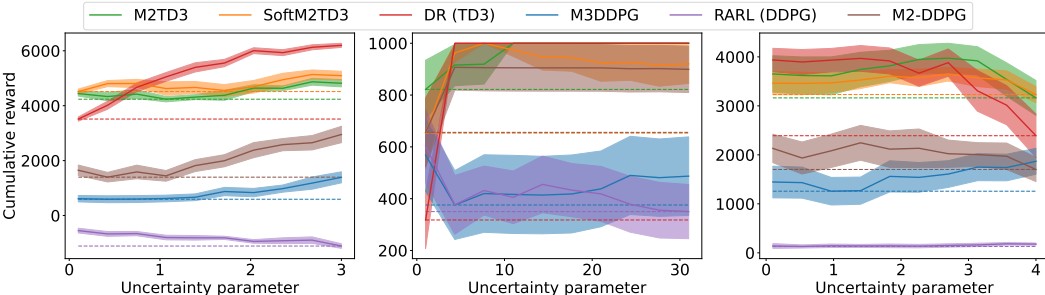

Figure 1: Cumulative rewards under trained policies for each uncertainty parameter $\omega \in \Omega$. The average (solid line) and standard error (band) for each $\omega \in \Omega$, as well as the worst average value (dashed line) are shown. Left: Ant 1, Middle: InvertedPendulum 1, Right: Walker 1.

**Experiment Setting**  We constructed 19 tasks based on six MuJoCo environments and with 1-3 uncertainty parameters, as summarized in Table 4 in Appendix C. Here, $\Omega$ was defined as an interval, which mostly included the default $\omega$ values.

To assess the worst-case performance of the given policy $\mu$ under $\omega \in \Omega$, we evaluated the cumulative reward 30 times for each uncertainty parameter value $\omega_1, \ldots, \omega_K \in \Omega$. Here, $R_k(\mu)$ was defined as the cumulative reward on $\omega_k$ averaged over 30 trials. Then, $R_{\mathrm{worst}}(\mu) = \min_{1 \leqslant k \leqslant K} R_k(\mu)$ was measured as an estimate of the worst-case performance of $\mu$ on $\Omega$. We also report the average performance $R_{\mathrm{average}}(\mu) = \frac{1}{K} \sum_{k=1}^{K} R_k(\mu)$. A total of $K$ uncertainty parameters $\omega_1, \ldots, \omega_K$ for evaluation were drawn as follows: for 1D $\omega$, we chose $K = 10$ equally spaced points on the 1D interval $\Omega$; for 2D $\omega$, we chose 10 equally spaced points in each dimension of $\Omega$, thereby obtaining $K = 100$ points; and for 3D $\omega$, we chose 10 equally spaced points in each dimension of $\Omega$, thereby obtaining $K = 1000$ points.

For each approach, we trained the policy 10 times in each environment. The training time steps $T_{\max}$ were set to 2M, 4M, and 5M for the senarios with 1D, 2D, and 3D uncertainty parameters, respectively. The final policies obtained were evaluated for their worst-case performances. For further details of the experiment settings, please refer to Appendix C.

**Comparison to Baseline Methods**  Table 1 summarizes the worst-case performances of the policies trained by M2TD3, SoftM2TD3, M2-DDPG, M3DDPG, RARL, and DR. See also Table 7 in Appendix E for the results of TD3 trained on the reference uncertainty parameters as baselines and Figure 2 in Appendix G for the learning curves. In most cases, M2TD3 and SoftM2TD3 outperformed DR, and RARL, and M3DDPG. Figure 1 shows the cumulative rewards of the policies trained by the

Table 2: Avg. $\pm$ std. error of average performance $R_{\mathrm{average}}(\mu)$ over 10 trials for each approach

| Environment | M2TD3 | SoftM2TD3 | M2-DDPG | M3DDPG | RARL (DDPG) | DR (TD3) |
|---|---|---|---|---|---|---|
| Ant 1 ($\times 10^3$) | $4.51 \pm 0.08$ | $4.78 \pm 0.16$ | $2.05 \pm 0.21$ | $0.84 \pm 0.14$ | $-0.82 \pm 0.06$ | $\mathbf{5.25 \pm 0.10}$ |
| Ant 2 ($\times 10^3$) | $5.44 \pm 0.05$ | $5.56 \pm 0.01$ | $3.03 \pm 0.19$ | $1.86 \pm 0.38$ | $-1.00 \pm 0.06$ | $\mathbf{6.32 \pm 0.09}$ |
| Ant 3 ($\times 10^3$) | $2.66 \pm 0.22$ | $2.98 \pm 0.23$ | $0.30 \pm 0.46$ | $-0.33 \pm 0.22$ | $-1.31 \pm 0.09$ | $\mathbf{3.62 \pm 0.11}$ |
| HalfCheetah 1 ($\times 10^3$) | $3.89 \pm 0.06$ | $4.00 \pm 0.05$ | $3.50 \pm 0.12$ | $1.01 \pm 0.26$ | $-0.46 \pm 0.02$ | $\mathbf{5.93 \pm 0.18}$ |
| HalfCheetah 2 ($\times 10^3$) | $4.35 \pm 0.05$ | $4.52 \pm 0.07$ | $3.91 \pm 0.08$ | $0.77 \pm 0.12$ | $-0.08 \pm 0.05$ | $\mathbf{5.79 \pm 0.15}$ |
| HalfCheetah 3 ($\times 10^3$) | $3.79 \pm 0.09$ | $4.02 \pm 0.04$ | $3.39 \pm 0.21$ | $0.58 \pm 0.18$ | $-0.21 \pm 0.10$ | $\mathbf{5.54 \pm 0.16}$ |
| Hopper 1 ($\times 10^3$) | $\mathbf{2.68 \pm 0.11}$ | $2.67 \pm 0.18$ | $1.79 \pm 0.22$ | $1.12 \pm 0.21$ | $0.38 \pm 0.08$ | $2.57 \pm 0.15$ |
| Hopper 2 ($\times 10^3$) | $\mathbf{2.51 \pm 0.07}$ | $2.26 \pm 0.12$ | $1.49 \pm 0.15$ | $1.15 \pm 0.11$ | $0.66 \pm 0.13$ | $1.89 \pm 0.08$ |
| Hopper 3 ($\times 10^3$) | $0.85 \pm 0.07$ | $0.79 \pm 0.04$ | $0.87 \pm 0.08$ | $0.49 \pm 0.11$ | $0.47 \pm 0.08$ | $\mathbf{1.50 \pm 0.07}$ |
| HumanoidStandup 1 ($\times 10^5$) | $1.08 \pm 0.04$ | $1.03 \pm 0.07$ | $1.05 \pm 0.06$ | $0.99 \pm 0.06$ | $0.77 \pm 0.06$ | $\mathbf{1.12 \pm 0.05}$ |
| HumanoidStandup 2 ($\times 10^5$) | $0.97 \pm 0.04$ | $\mathbf{1.07 \pm 0.05}$ | $0.93 \pm 0.04$ | $0.92 \pm 0.04$ | $0.85 \pm 0.08$ | $1.06 \pm 0.04$ |
| HumanoidStandup 3 ($\times 10^5$) | $\mathbf{1.09 \pm 0.06}$ | $1.04 \pm 0.03$ | $0.98 \pm 0.06$ | $1.01 \pm 0.04$ | $0.87 \pm 0.07$ | $1.04 \pm 0.07$ |
| InvertedPendulum 1 ($\times 10^2$) | $\mathbf{9.66 \pm 0.25}$ | $9.17 \pm 0.54$ | $8.79 \pm 0.87$ | $4.51 \pm 0.12$ | $4.21 \pm 0.81$ | $9.32 \pm 0.11$ |
| InvertedPendulum 2 ($\times 10^2$) | $6.13 \pm 1.42$ | $6.26 \pm 0.95$ | $8.27 \pm 0.70$ | $1.76 \pm 0.51$ | $3.07 \pm 0.66$ | $\mathbf{9.18 \pm 0.07}$ |
| Walker 1 ($\times 10^3$) | $\mathbf{3.70 \pm 0.31}$ | $3.51 \pm 0.16$ | $2.03 \pm 0.26$ | $1.55 \pm 0.25$ | $0.15 \pm 0.03$ | $3.59 \pm 0.26$ |
| Walker 2 ($\times 10^3$) | $\mathbf{4.72 \pm 0.12}$ | $4.37 \pm 0.32$ | $2.39 \pm 0.20$ | $1.63 \pm 0.22$ | $0.26 \pm 0.05$ | $4.54 \pm 0.31$ |
| Walker 3 ($\times 10^3$) | $4.27 \pm 0.21$ | $4.21 \pm 0.30$ | $2.48 \pm 0.24$ | $1.65 \pm 0.15$ | $0.21 \pm 0.07$ | $\mathbf{4.48 \pm 0.16}$ |
| Small HalfCheetah 1 ($\times 10^3$) | $6.00 \pm 0.15$ | $6.04 \pm 0.13$ | $5.71 \pm 0.07$ | $3.38 \pm 0.34$ | $-0.42 \pm 0.01$ | $\mathbf{8.11 \pm 0.17}$ |
| Small Hopper 1 ($\times 10^3$) | $2.96 \pm 0.31$ | $2.44 \pm 0.31$ | $1.57 \pm 0.21$ | $1.53 \pm 0.21$ | $0.56 \pm 0.13$ | $\mathbf{3.44 \pm 0.10}$ |

Table 3: Avg. $\pm$ std. error of worst-case performance $R_{\mathrm{worst}}(\mu)$ and average performance $R_{\mathrm{average}}(\mu)$ on InvertedPendulum 1 obtained by M2TD3 variants over 10 trials

| Environment | N=5 | N=1 | N=10 | w/o DRS | w/o PRS |
|---|---|---|---|---|---|
| worst-case ($\times 10^2$) | $8.22 \pm 1.13$ | $5.77 \pm 1.29$ | $9.07 \pm 0.89$ | $9.07 \pm 0.88$ | $5.41 \pm 1.45$ |
| average ($\times 10^2$) | $9.66 \pm 0.25$ | $8.10 \pm 0.90$ | $9.89 \pm 0.11$ | $9.08 \pm 0.88$ | $6.90 \pm 1.25$ |

six approaches for each of the $\omega$ values on the Ant 1, InvertedPendulum 1, and Walker 1 senarios. See also Figure 4 in Appendix H for other senarios.

Comparison with DR: DR does not attempt to optimize the worst-case performance. In fact, it showed lower worst-case performance than M2TD3 and SoftM2TD3 in many scenarios because the obtained policy performs poorly on some uncertainty parameters while it performs well on average. In the results of InvertedPendulum 1 shown in Figure 1, for example, the policy obtained by DR exhibited a high performance for a wide range of $\omega \in \Omega$ but performed poorly for small $\omega$. M2TD3 outperformed DR and the other baselines on those senarios. However, in some scenarios, such as HumanoidStandup 1, Small HalfCheetah 1, and Small Hopper 1, DR achieved a better worst-case performance than M2TD3. This outcome may be because the optimization of worst-case function values is generally more unstable than that of the expected function values.

Comparison with RARL: M2-DDPG outperformed RARL in most senarios, with this performance difference originating from the different optimization strategies. As noted above, the optimization strategy employed in RARL often fails to converge. Specifically, as shown in Figure 1, RARL failed to optimize the policy not only in the worst-case but also on average in Ant 1 and Walker 1 senarios. In the InvertedPendulum 1 senario, RARL could train the policy for some uncertainty parameters, but not for the worst-case uncertainty parameter.

Comparison with M3DDPG: In many cases, M2-DDPG outperformed M3DDPG. Because M3DDPG considers a stronger adversary than necessary (the worst $\omega$ for each time step), it was too conservative for this experiment and exhibited lower performance.

**Average Performance** Although our objective is maximizing the worst-case performance, the average performance is also important in practice. Table 2 compares the average performance of the six approaches. Generally, DR achieved the highest average performance as expected. However, interestingly, M2TD3 and SoftM2TD3 achieved competitive average performances to DR on several senarios such as Hopper 1, 2, and Walker 1, 2. Moreover, Table 1 compared with Table 2 shows that a few times higher average performance than worst-case performance on several senarios such as Ant 3 and Hopper 1–3.

**Ablation Study**    Ablation studies were conducted on InvertedPendulum 1 with the pole mass as the uncertainty parameters. M2TD3 with $N = 5$ was taken as the baseline. We tested several variants as follows: with different multiple uncertainty parameters ($N = 1$ and 10), without a distance-based refreshing strategy (w/o DRS), and without a probability-based refreshing strategy (w/o PRS). Table 3 shows the results. The inclusion of multiple uncertainty parameters and the probability-based refresh strategy contributed significantly to the worst-case performance and average performance, implying that both techniques contribute to better estimating the worst uncertainty parameter. Although DRS had little impact on the performance, prior knowledge in DRS can be implemented by defining the distance and the threshold in the uncertainty parameter task-dependently, which we did not implement here.

**Small Uncertainty Set (Limitation of M2TD3)**    Small HalfCheetah 1 and Small Hopper 1 were designed to have a smaller uncertainty parameter set than HalfCheetah 1 and Hopper 1 and to reveal the effect of the size of the uncertainty parameter interval. As expected, a smaller uncertainty parameter set resulted in higher worst-case (Table 1) and average (Table 2) performances, for all approaches. Because M2TD3 performs the worst-case optimization, it is expected to show better worst-case performance than DR, independently of the size of the uncertainty parameter set. However, in these senarios, DR showed better worst-case performance than that of M2TD3, while M2TD3 achieved competitive or superior worst-case performance to those of DR on HalfCheetah 1 and Hopper 1. This may be because the max-min optimization performed by M2TD3 resulted in sub-optimal policies. When the uncertainty parameter set is small, and the performance does not change significantly over the uncertainty parameter set, maximizing the average performance is likely to lead to high worst-case performance. The sub-optimal results obtained by M2TD3 is then dominated by the results obtained by DR. Therefore, the maxmin optimization in M2TD3 can be improed.

**Evaluation Under Adversarial External Force**    Although we developed the proposed approach for the situation that the uncertainty parameter is directly encoded by $\omega$, we can extend it to the situation where the model misspecification is expressed by an external force produced by an adversarial agent as in [31]. The extended approach and its experimental result are given in Appendix J. Superior worst-case performances of M2TD3 over DR, RARL, and M3DDPG were observed for this setting as well.

## 6    Conclusion

In this study, we targeted the policy optimization aimed at maximizing the worst-case performance in a predefined uncertainty set. The list of the contributions are as follows. (i) We formulated the off-policy deterministic actor-critic approach to the worst-case performance maximization as a tri-level optimization problem (3). (ii) We developed the worst-case performance of M2TD3. The key concepts were the incorporation of the uncertainty parameter into the critic network and use of the simultaneous gradient ascent descent method. Different technical components were introduced to stabilize the training. (iii) We evaluated the worst-case performance of M2TD3 on 19 MuJoCo tasks through comparison with three baseline methods. Ablation studies revealed the usefulness of each component of M2TD3.

## Acknowledgements

This research is partially supported by the JSPS KAKENHI Grant Number 19H04179 and the NEDO Project Number JPNP18002.

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
