# A  Algorithm Details

Algorithm 2 summarizes the overall framework of M2TD3. The training is stalled if the size of the replay buffer is smaller than the minibatch size, i.e., if $|B| < M$. Algorithms 3 and 4 show the critic network update and the actor network and uncertainty parameter sampler update, respectively. Although we write the gradient-based update in the form of a mini-batch stochastic gradient update for simplicity, we employ an adaptive approach such as Adam [16].

---

**Algorithm 2** M2TD3 (Framework)

---

1: `# initialization`
2: Initialize uncertainty parameters $\hat{\omega}_i \in \mathcal{U}(\Omega)$ for $i = 1, \ldots, N$
3: Initialize policy parameter $\theta$ and critic parameters $\phi_1, \phi_2$ with random values
4: Initialize target network parameters $\theta' \leftarrow \theta, \phi'_1 \leftarrow \phi_1, \phi'_2 \leftarrow \phi_2$
5: Initialize frequency parameter $p_1, \ldots, p_N = 1/N$
6: Initialize replay buffer $B = \emptyset$
7: `# training loop`
8: Draw uncertainty parameter $\omega \sim \alpha_0$
9: Observe initial state $s \sim p^0_\omega$
10: **for** t = 1 to $T_{\max}$ **do**
11:    `# interaction`
12:    Select action $a \sim \beta_t(s)$
13:    Interact with $\mathcal{M}_\omega$ with $a$, observe next state $s'$, immediate reward $r$ and termination flag $h$
14:    Store transition tuple $(s, a, r, s', h, \omega)$ in $B$
15:    **if** $h = 1$ **then**
16:       Reset uncertainty parameter $\omega \sim \alpha_t$
17:       Observe initial state $s \sim p^0_\omega$
18:    **else**
19:       Update current state $s \leftarrow s'$
20:    **end if**
21:    `# learning`
22:    Sample mini-batch $\{(s_i, a_i, r_i, s'_i, h_i, \omega_i)\}$ of M transitions uniform-randomly from $B$
23:    Perform Algorithm 3 for critic network update
24:    **if** $\mathrm{mod}(t, T_{\mathrm{freq}}) = 0$ **then**
25:       Perform Algorithm 4 for actor network update and uncertainty parameter sampler update
26:       Update target networks analogously to TD3
27:    **end if**
28: **end for**

---

---

**Algorithm 3** Critic Update

---

1: **for** $i = 1, \ldots, M$ **do**
2:    $\tilde{a}_i \leftarrow \mu_{\theta'}(s'_i) + \epsilon_a, \epsilon_a \sim \Pi_a(\mathcal{N}(0, \tilde{\Sigma}_a))$
3:    $\tilde{\omega}'_i \leftarrow \omega_i + \epsilon_\omega, \epsilon_\omega \sim \Pi_\omega(\mathcal{N}(0, \tilde{\Sigma}_\omega))$
4:    $y_i \leftarrow r_i + \min\{Q_{\phi'_1}(s'_i, \tilde{a}_i, \tilde{\omega}_i), Q_{\phi'_2}(s'_i, \tilde{a}_i, \tilde{\omega}_i)\}$
5: **end for**
6: **for** $j = 1, 2$ **do**
7:    $\phi_j \leftarrow \phi_j - \lambda_\phi \nabla_{\phi_j} \frac{1}{M} \sum_{i=1}^{M} (y_i - Q_{\phi_j}(s_i, a_i, \omega_i))^2$
8: **end for**

---

We maintain the frequency $p_k$ of each uncertainty parameter $\hat{\omega}_k$ being the worst one among $\hat{\omega}_1, \ldots, \hat{\omega}_N$ in Algorithm 4. This is used in two ways: criteria for the refreshing strategy of $\hat{\omega}_k$ in Algorithm 4; and mixture weights for the uncertainty parameter sampler $\alpha$. The update of $p_k$ follows the exponential moving average with the momentum $(1/T_{\mathrm{last}})$, where $T_{\mathrm{last}}$ is the number of steps spent in the last episode ($T_{\mathrm{last}}$ is set to 1000 for the first episode). The reason behind this design choice is as follows. The short episode is a meaning that a bad uncertainty parameter $\omega$ is used in the last episode. Because the uncertainty parameter $\omega$ used in the interaction is sampled from $\alpha$, which is a gaussian mixture with components centered at $\hat{\omega}_1, \ldots, \hat{\omega}_k$, it implies that they include a bad uncertainty parameter as well. Then, there is a high chance that this uncertainty parameter is selected

**Algorithm 4** Actor Update and Uncertainty Parameter Sampler Update

---

1: # maximin update
2: $k' = \operatorname{argmin}_k \frac{1}{M} \sum_{i=1}^{M} Q_{\phi^1}(s_i, \mu_\theta(s_i), \hat{\omega}_k)$
3: $\theta \leftarrow \theta + \lambda_\theta \nabla_\theta \frac{1}{M} \sum_{i=1}^{M} Q_{\phi^1}(s_i, \mu_\theta(s_i), \hat{\omega}_{k'})$
4: $\hat{\omega}_{k'} \leftarrow \hat{\omega}_{k'} - \lambda_\omega \nabla_{\hat{\omega}_{k'}} \frac{1}{M} \sum_{i=1}^{M} Q_{\phi^1}(s_i, \mu_\theta(s_i), \hat{\omega}_{k'})$
5: # uncertainty parameter sampler update with refreshing strategy
6: **for** $k = 1, \ldots, N$ **do**
7: $\quad \hat{\omega}_k \sim \mathcal{U}(\Omega)$ **if** $d_\omega(\hat{\omega}_k, \hat{\omega}_\ell) \leqslant d_{\text{thre}}$ for some $\ell \neq k$
8: $\quad \hat{\omega}_k \sim \mathcal{U}(\Omega)$ **if** $p_k \leqslant p_{\text{thre}}$
9: $\quad$ **if** $\hat{\omega}_k$ is refreshed **then**
10: $\quad\quad p_k \leftarrow 1/N$
11: $\quad$ **else**
12: $\quad\quad p_k \leftarrow (1 - 1/T_{\text{last}})p_k + (1/T_{\text{last}})\mathbb{I}\{k = k'\}$
13: $\quad$ **end if**
14: **end for**
15: $p_k \leftarrow p_k / \sum_{\ell=1}^{n} p_\ell$ for all $k = 1, \ldots, N$

---

as the worst uncertainty parameter $\hat{\omega}_{k'}$ in Algorithm 4. By setting a greater momentum $(1/T_{\text{last}})$, we can accelerate the approach of $p_k$ to $\mathbb{I}\{k = k'\}$, which is 1 if $k = k'$ and 0 otherwise. With this fast update of $p_k$, we expect two consequences. First, there are increased chances to sample $\omega$ around the worst $\hat{\omega}_{k'}$. Second, there are increased chances to refresh the non-worst uncertainty parameters, which leads to more exploration of the worst uncertainty parameter search. Preliminary experiments have confirmed that this momentum setting leads to a better worst-case performance than a constant momentum.

Our implementation is publicly available (https://github.com/akimotolab/M2TD3).

## B  Soft-Min Variant: SoftM2TD3

In theory, M2TD3 can obtain a policy that exhibits a better worst-case performance than a policy obtained by DR as DR does not explicitly maximize the worst-case performance. However, we empirically observe that DR sometimes achieves a better worst-case performance than M2TD3 on senarios where the performance does not change significantly over the uncertainty set $\Omega$ (Table 1). We conjecture that this is because the difficulty in the max-min optimization of M2TD3 compared to the optimization of the expectation in DR.

To mitigate this issue, we propose a variant of M2TD3, called SoftM2TD3. The objective of the update of the policy parameter $\theta$ is replaced with the following soft-min version:

$$\tilde{J}_t(\theta) = \sum_{k=1}^{N} w_k \left[ \frac{1}{M} \sum_{i=1}^{M} Q_{\phi_t}(s_i, \mu_\theta(s), \hat{\omega}_k) \right] , \tag{8}$$

where $w_k$ is the weight for uncertainty parameter $\hat{\omega}_k$. A greater weight value should be assigned to $\hat{\omega}_k$ with smaller Q-values. We used the frequency $p_k$ of $\hat{\omega}_k$ as the worst-case during the actor update, i.e., $w_k = p_k$. This update is close to M2TD3 if $p = (p_1, \ldots, p_N)$ is close to a one-hot vector and is close to DR if $p_1 \approx \cdots \approx p_N$, which is the case at the beginning of the training.

The objective function of SoftM2TD3 is considered an approximation of the objective function of M2TD3. The difference between M2TD3 and SoftM2TD3 is in the optimization process. Because the objective function of SoftM2TD3 is closer to that of DR, we expect that the optimization in SoftM2TD3 is more efficient than M2TD3, where the effect of the accuracy of the estimation of the worst-case uncertainty parameters is greater than SoftM2TD3.

## C  Experiment Details

Table 4 lists the senario we used in our experiments. These senarios are created by setting 1 to 3 constants in the original MuJoCo environment to the uncertainty parameters. The uncertainty set

Table 4: List of senarios used in the experiments

| Environment | Uncertainty Set $\Omega$ | Reference Parameter | Uncertainty Parameter Name |
|---|---|---|---|
| Baseline MuJoCo Environment: Ant | | | |
| Ant 1 | [0.1, 3.0] | 0.33 | torso mass |
| Ant 2 | [0.1, 3.0] $\times$ [0.01, 3.0] | (0.33, 0.04) | torso mass $\times$ front left leg mass |
| Ant 3 | [0.1, 3.0] $\times$ [0.01, 3.0] $\times$ [0.01, 3.0] | (0.33, 0.04, 0.06) | torso mass $\times$ front left leg mass $\times$ front right leg mass |
| Baseline MuJoCo Environment: HalfCheetah | | | |
| HalfCheetah 1 | [0.1, 4.0] | 0.4 | world friction |
| HalfCheetah 2 | [0.1, 4.0] $\times$ [0.1, 7.0] | (0.4, 6.36) | world friction $\times$ torso mass |
| HalfCheetah 3 | [0.1, 4.0] $\times$ [0.1, 7.0] $\times$ [0.1, 3.0] | (0.4, 6.36, 1.53) | world friction $\times$ torso mass $\times$ back thigh mass |
| Baseline MuJoCo Environment: Hopper | | | |
| Hopper 1 | [0.1, 3.0] | 1.00 | world friction |
| Hopper 2 | [0.1, 3.0] $\times$ [0.1, 3.0] | (1.00, 3.53) | world friction $\times$ torso mass |
| Hopper 3 | [0.1, 3.0] $\times$ [0.1, 3.0] $\times$ [0.1, 4.0] | (1.00, 3.53, 3.93) | world friction $\times$ torso mass $\times$ thigh mass |
| Baseline MuJoCo Environment: HumanoidStandup | | | |
| HumanoidStandup 1 | [0.1, 16.0] | 8.32 | torso mass |
| HumanoidStandup 2 | [0.1, 16.0] $\times$ [0.1, 8.0] | (8.32, 1.77) | torso mass $\times$ right foot mass |
| HumanoidStandup 3 | [0.1, 16.0] $\times$ [0.1, 5.0] $\times$ [0.1, 8.0] | (8.32, 1.77, 4.53) | torso mass $\times$ right foot mass $\times$ left thigh mass |
| Baseline MuJoCo Environment: Inveted Pendulum | | | |
| InvertedPendulum 1 | [1.0, 31.0] | 4.90 | pole mass |
| InvertedPendulum 2 | [1.0, 31.0] $\times$ [1.0, 11.0] | (4.90, 9.42) | pole mass $\times$ cart mass |
| Baseline MuJoCo Environment: Walker | | | |
| Walker 1 | [0.1, 4.0] | 0.7 | world friction |
| Walker 2 | [0.1, 4.0] $\times$ [0.1, 5.0] | (0.7, 3.53) | world friction $\times$ torso mass |
| Walker 3 | [0.1, 4.0] $\times$ [0.1, 5.0] $\times$ [0.1, 6.0] | (0.7, 3.53, 3.93) | world friction $\times$ torso mass $\times$ thigh mass |
| Small MuJoCo Environments | | | |
| Small HalfCheetah 1 | [0.1, 3.0] | 0.4 | world friction |
| Small Hopper 1 | [0.1, 2.0] | 1.00 | world friction |

is designed as an interval. Except for Hopper 2 and Hopper 3, the original value is included in the uncertainty set; hence, the trivial upper bound of the worst-case performance is the best performance of the corresponding MuJoCo environment.

In all approaches in all senarios, the same configurations are used for fair comparison as follows.

For DR and RARL, the input to the critic is a $(s, a)$ pair, whereas it is a tuple of $(s, a, \omega)$ for M2TD3, SoftM2TD3, and M3DDPG. Except for this point, we used the same network architecture. The policy and the critic networks are defined as fully connected layers with two hidden layers of size 256.

The uncertainty parameter sampler $\alpha_t$ is

$$\alpha_t = \begin{cases} \mathcal{U}(\Omega) & t \leqslant T_{\text{rand}} \\ \Pi_\Omega(\sum_{k=1}^N p_k \cdot \mathcal{N}(\hat{\omega}_k, \Sigma_\omega)) & t > T_{\text{rand}} \end{cases}, \tag{9}$$

where $\Sigma_\omega$ is a diagonal matrix, with diagonal elements 0.5 times the lengths of the intervals of the corresponding dimension of $\Omega$, and $\Pi_\Omega$ is the projection onto $\Omega$ and $T_{\text{rand}} = 10^5$. The behavior policy $\beta_t$ is

$$\beta_t = \begin{cases} \mathcal{U}(A) & t \leqslant T_{\text{rand}} \\ \Pi_A(\mathcal{N}(\mu_\theta(s), \Sigma_a)) & t > T_{\text{rand}} \end{cases}, \tag{10}$$

where $\Sigma_a$ is a diagonal matrix, with diagonal elements are the 0.5 times the length of the intervals of the corresponding dimension of $A$, and $\Pi_A$ is the projection onto $A$. The $\Sigma_\omega$ element is designed to decay at each time step to 0.05 times the length of the intervals of it when the learning progresses to half of the total.

The minibatch size is $M = 100$. The learning rates for the actor update, the uncertainty parameter update and the critic update are $\lambda_\theta = \lambda_\omega = \lambda_\phi = 3 \times 10^{-4}$. The noise covariance matrices for the target policy smoothing are $\tilde{\Sigma}_a = 2\Sigma_a$ and $\tilde{\Sigma}_\omega = 2\Sigma_\omega$. The noise for the target policy smoothing is clipped by $\Pi_a$ and $\Pi_\omega$ into the ranges of $\pm 0.25$ times the interval lengths of the corresponding

Table 5: Taxonomy of robust policy search. Cont.: Continuous space. Disc.: Discrete space.

| Method | Uncertainty | $S$ & $A$ | $\Omega$ |
|---|---|---|---|
| R-MPO | Transition | Cont. | Disc. |
| ROPI | Transition | Disc. | Cont. |
| ATLA, SA-DRL | Observation | Cont. | Cont. |
| PAIRED, RARL, M3DDPG RRL, Adv-SAC, RSAC, M2TD3 | Reward & Transition | Cont. | Cont. |

Table 6: Comparison of action value functions in related methods

| Method | Action Value Function |
|---|---|
| R-MPO, ROPI, Robust DP | $Q(s,a) = r(s,a) + \gamma \min_{\omega \in \Omega} \mathbb{E}[Q(s', \mu_\theta(s')) \mid s' \sim p_\omega(\cdot \mid s,a)]$ |
| DR | $Q(s,a) = \mathbb{E}_{\omega \in \Omega}[r_\omega(s,a) + \gamma \mathbb{E}[Q(s', \mu_\theta(s')) \mid s' \sim p_\omega(\cdot \mid s,a)]]$ |
| RRL, RARL, Adv-SAC | $Q_\omega(s,a) = r_\omega(s,a) + \gamma \mathbb{E}[Q_\omega(s', \mu_\theta(s')) \mid s' \sim p_\omega(\cdot \mid s,a)]$ |
| M3DDPG | $Q(s,a,\omega) = r_\omega(s,a) + \gamma \mathbb{E}[\min_{\omega' \in \Omega} Q(s', \mu_\theta(s'), \omega') \mid s' \sim p_\omega(\cdot \mid s,a)]$ |
| M2TD3 | $Q(s,a,\omega) = r_\omega(s,a) + \gamma \mathbb{E}[Q(s', \mu_\theta(s'), \omega) \mid s' \sim p_\omega(\cdot \mid s,a)]$ |

dimensions of $A$ and $\Omega$, respectively. The actor and target network update frequency is $T_{\text{freq}} = 2$. The above parameter settings follow TD3 [7].

We used $N = 5$ uncertainty parameters. For the refreshing strategy of the uncertainty parameters, we used $\ell_1$-distance as $d_\omega$. The distance threshold is $d_{\text{thre}} = 0.1$. The frequency threshold is $0.05$.

Experiments were performed on a machine with two NVIDIA RTX A5000 GPUs, two Intel(R) Xeon(R) Gold 6230 CPUs, and 192GB memory.

# D  Related Work

Table 5 summarizes the robust policy search method taxonomy. Table 6 compares the related methods considering their action value function definitions.

# E  Performance of TD3

Table 7 summarizes the worst-case performance and the average performance of the policies obtained by TD3 on the Ant, HalfCheetah, Hopper, HumanoidStandup, InvertedPendulum, and Walker environments with their reference parameters. The results show that the original TD3 policies learned under the reference parameter cannot be generalized to uncertain parameters set in most scenarios. These results provide the baseline performances on these environments and show that these environments are non-trivial for the worst-case and the average performance maximization.

# F  Original RARL

The performance of the original RARL, whose baseline RL approach is TRPO, is shown in Table 8. RARL (TRPO) exhibited low worst-case performances, similarly to those of RARL (DDPG). These low performances of RARL (TRPO) may be due to its defect of the optimization strategy and because TRPO is an on-policy method, which often requires a greater number of interactions than off-policy methods [10].

# G  Learning Curve

The learning curves for each approach are shown in the Figure 2 and Figure 3. Figure 2 shows that M2TD3 and SoftM2TD3 tended to perform better in the early stages of learning than DR, even in scenarios where the final worst-case performances of DR, M2TD3, and SoftM2TD3 were competitive.

Table 7: Avg. $\pm$ std. error of worst-case performance $R_{\mathrm{worst}}(\mu)$ and average performance $R_{\mathrm{average}}(\mu)$ and reference performance $R_{\mathrm{ref}}(\mu)$ over 10 trials for TD3 (reference parameter)

| Environment | worst | average |
|---|---|---|
| Ant (reference parameters): $3.02 \pm 0.15 (\times 10^3)$ | | |
| Ant 1 $(\times 10^3)$ | $2.22 \pm 0.50$ | $2.76 \pm 0.50$ |
| Ant 2 $(\times 10^3)$ | $1.59 \pm 0.08$ | $2.28 \pm 0.09$ |
| Ant 3 $(\times 10^2)$ | $-0.99 \pm 1.13$ | $3.16 \pm 1.00$ |
| HalfCheetah (reference parameters): $10.2 \pm 0.2 (\times 10^3)$ | | |
| Small HalfCheetah 1 $(\times 10^3)$ | $0.03 \pm 0.11$ | $3.73 \pm 0.29$ |
| HalfCheetah 1 $(\times 10^3)$ | $-0.34 \pm 0.04$ | $2.79 \pm 0.22$ |
| HalfCheetah 2 $(\times 10^3)$ | $-0.53 \pm 0.06$ | $2.63 \pm 0.20$ |
| HalfCheetah 3 $(\times 10^3)$ | $-0.61 \pm 0.08$ | $2.47 \pm 0.18$ |
| Hopper (reference parameters): $3.01 \pm 0.19 (\times 10^3)$ | | |
| Small Hopper 1 $(\times 10^3)$ | $2.84 \pm 0.22$ | $2.97 \pm 0.21$ |
| Hopper 1 $(\times 10^3)$ | $0.40 \pm 0.02$ | $2.39 \pm 0.14$ |
| Hopper 2 $(\times 10^3)$ | $0.21 \pm 0.04$ | $1.54 \pm 0.17$ |
| Hopper 3 $(\times 10^3)$ | $0.14 \pm 0.03$ | $1.15 \pm 0.14$ |
| HumanoidStandup (reference parameters): $1.08 \pm 0.03 (\times 10^5)$ | | |
| HumanoidStandup 1 $(\times 10^5)$ | $0.85 \pm 0.07$ | $1.03 \pm 0.04$ |
| HumanoidStandup 2 $(\times 10^5)$ | $0.73 \pm 0.07$ | $1.03 \pm 0.03$ |
| HumanoidStandup 3 $(\times 10^5)$ | $0.57 \pm 0.04$ | $1.01 \pm 0.03$ |
| InvertedPendulum (reference parameters): $10.0 \pm 0.0 (\times 10^2)$ | | |
| InvertedPendulum 1 $(\times 10^2)$ | $0.24 \pm 0.10$ | $7.34 \pm 0.76$ |
| InvertedPendulum 2 $(\times 10^2)$ | $0.03 \pm 0.00$ | $4.05 \pm 0.52$ |
| Walker (reference parameters): $4.08 \pm 0.16 (\times 10^3)$ | | |
| Walker 1 $(\times 10^3)$ | $0.68 \pm 0.12$ | $3.12 \pm 0.20$ |
| Walker 2 $(\times 10^3)$ | $0.28 \pm 0.07$ | $2.70 \pm 0.20$ |
| Walker 3 $(\times 10^3)$ | $0.17 \pm 0.06$ | $2.60 \pm 0.18$ |

Table 8: Avg. $\pm$ std. error of worst-case performance $R_{\mathrm{worst}}(\mu)$ and average performance $R_{\mathrm{average}}(\mu)$ over 10 trials for RARL (TRPO)

| Environment | worst | average |
|---|---|---|
| Ant 1 $(\times 10^1)$ | $-4.92 \pm 0.48$ | $-2.29 \pm 0.22$ |
| Ant 2 $(\times 10^2)$ | $-1.15 \pm 0.19$ | $-0.37 \pm 0.04$ |
| Ant 3 $(\times 10^2)$ | $-0.32 \pm 0.41$ | $1.28 \pm 0.68$ |
| HalfCheetah 1 $(\times 10^2)$ | $-3.06 \pm 0.85$ | $1.22 \pm 0.63$ |
| HalfCheetah 2 $(\times 10^2)$ | $-5.23 \pm 0.89$ | $-0.03 \pm 0.73$ |
| HalfCheetah 3 $(\times 10^2)$ | $-9.70 \pm 1.89$ | $-0.70 \pm 0.65$ |
| Hopper 1 $(\times 10^2)$ | $2.89 \pm 0.25$ | $3.49 \pm 0.38$ |
| Hopper 2 $(\times 10^2)$ | $2.67 \pm 0.26$ | $4.82 \pm 0.64$ |
| Hopper 3 $(\times 10^2)$ | $0.71 \pm 0.13$ | $2.24 \pm 0.47$ |
| HumanoidStandup 1 $(\times 10^4)$ | $5.30 \pm 0.22$ | $6.78 \pm 0.21$ |
| HumanoidStandup 2 $(\times 10^4)$ | $5.04 \pm 0.09$ | $6.99 \pm 0.25$ |
| HumanoidStandup 3 $(\times 10^4)$ | $4.99 \pm 0.08$ | $6.64 \pm 0.17$ |
| InvertedPendulum 1 $(\times 10^2)$ | $0.44 \pm 0.10$ | $3.03 \pm 0.98$ |
| InvertedPendulum 2 $(\times 10^2)$ | $0.16 \pm 0.05$ | $5.86 \pm 0.87$ |
| Walker 1 $(\times 10^2)$ | $2.96 \pm 0.16$ | $3.79 \pm 0.34$ |
| Walker 2 $(\times 10^2)$ | $2.53 \pm 0.22$ | $4.16 \pm 0.34$ |
| Walker 3 $(\times 10^2)$ | $2.82 \pm 0.17$ | $3.90 \pm 0.14$ |
| Small HalfCheetah 1 $(\times 10^2)$ | $-0.89 \pm 0.80$ | $3.44 \pm 0.95$ |
| Small Hopper 1 $(\times 10^2)$ | $4.37 \pm 0.40$ | $4.44 \pm 0.42$ |

## H  Cumulative Rewards Under Different Uncertainty Parameters

Figures 4 to 6 show the cumulative rewards of the policies trained by the seven approaches for each $\omega \in \Omega$. These results show that in many scenarios, DR achieved good performances in a wide range of uncertainty parameters. However, the differences between the worst-case and best-case performances were relatively large for DR, and the worst-case performances were relatively low. Unlike DR, M2TD3 exhibited smaller performance differences between the worst-case and the best-case, and its worst-case performances were relatively high.

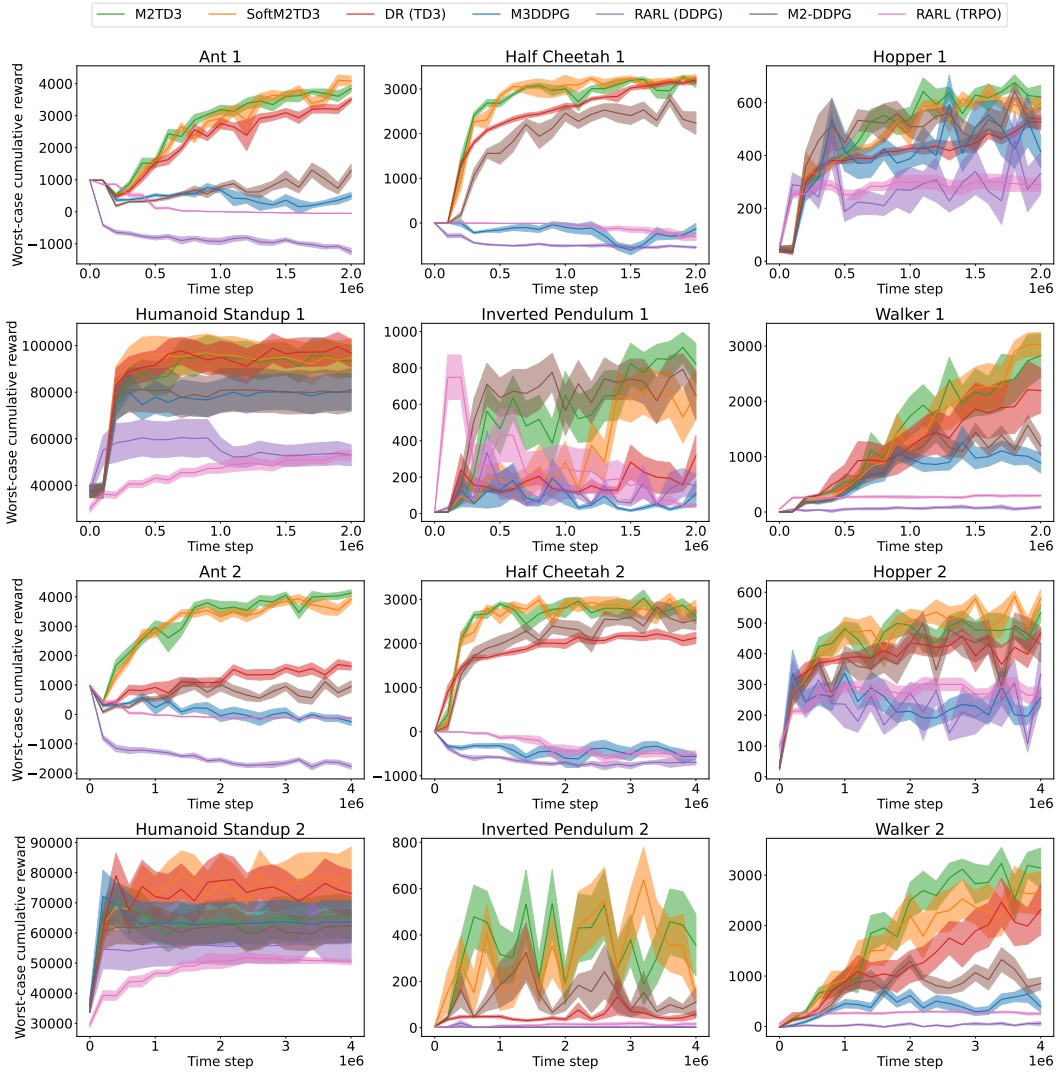

Figure 2: Learning curve for worst-case performance: the average (solid line) and standard error (band) of the worst-case cumulative rewards $R_{\text{worst}}(\mu)$

# I  Multiple Uncertainty Parameter

Figure 7 shows the learning curves for each uncertainty parameter $\omega \in \Omega$ and the uncertainty parameter $\hat{\omega}$ used to update the actor ($\hat{\omega}_{k'}$ in Algorithm 4) at each time step during the training. For each senario, the cumulative rewards under 10 equally spaced uncertainty parameters were evaluated every $1e5$ time steps. Hence, the lowest cumulative reward at each time step is a rough estimate of the worst-case performance. The uncertainty parameter with the lowest Q-value, $\operatorname{argmin}_{\hat{\omega}_1,\dots,\hat{\omega}_k,\dots\hat{\omega}_N} J_t(\theta, \hat{\omega}_k)$, is chosen for the update of the actor among $N$ worst uncertainty parameter candidates. This means that the selected worst-case uncertainty parameter can be different from the ground truth worst-case uncertainty parameter because of incomplete optimization for the worst uncertainty parameter, incomplete training of the critic network, and discrepancy between the cumulative reward and the Q-values.

Focusing on the behavior of M2TD3 on the HalfCheetah 1 senario (left-most side figures), the cumulative reward (top figure) shows that the uncertainty parameters $0.1$ (green line), $0.5$ (orange line), and $4.0$ (light blue line) alternately came to the bottom during learning, indicating that the worst uncertainty parameter continues to change between these values. The uncertainty parameter selected during the actor training (bottom figure) were the values around $0.1$ and $4.0$ in the early stages of the

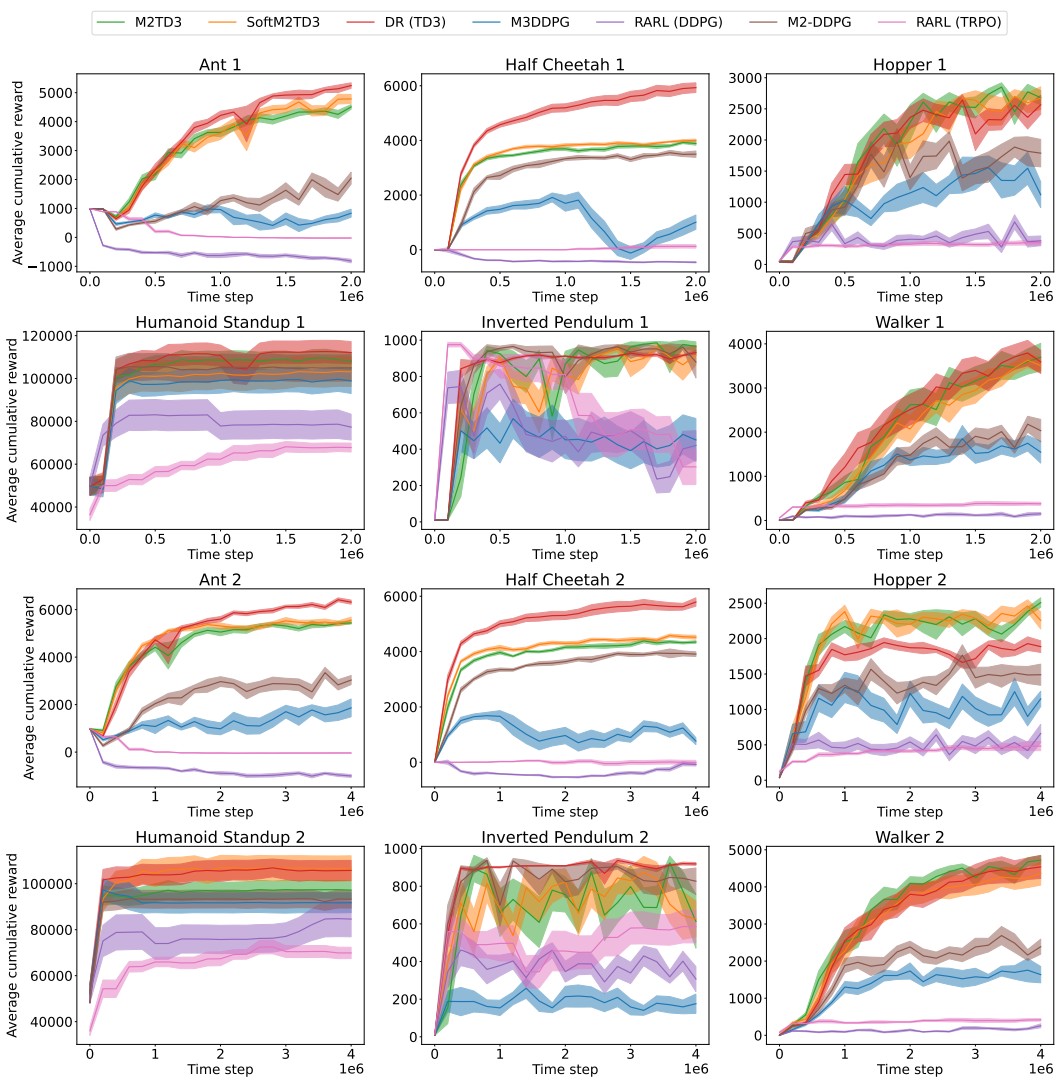

Figure 3: Learning curve for average performance: the average (solid line) and standard error (band) of the average cumulative rewards $R_{\text{average}}(\mu)$

training, and the values around $0.5$ were used in the middle of the training. This indicates that the algorithm was able to track the change of the worst-case uncertainty parameter during the training. This behavior of M2TD3 and SoftM2TD3 is shown on the other tasks.

## J Evaluation Under Adversarial External Force

With a small modification of the proposed approaches, we can apply it to the situation conforming to [31], where the model misspecification is expressed by an external force given by an adversarial agent. In this section, we describe the necessary modification of the proposed approaches and compare the worst-case performance with baselines.

**Extension of M2TD3** In this setting, we deal with an MDP $\mathcal{M} = \langle S, A_p, A_a, p, p_0, r, \gamma \rangle$, where $A_p$ and $A_a$ are the action spaces of the protagonist agent and adversarial agent, respectively, and they are assumed to be continuous. The transition probability density $p : S \times A_p \times A_a \times S \to \mathbb{R}$ and immediate reward $r : S \times A_p \times A_a \to \mathbb{R}$ are not anymore parameterized by $\omega$ but take the action of the adversarial agent as input. Unless otherwise specified, the other notations are the same as the ones in Section 2. The protagonist and adversarial agents interact with the environment $\mathcal{M}$ using stochastic

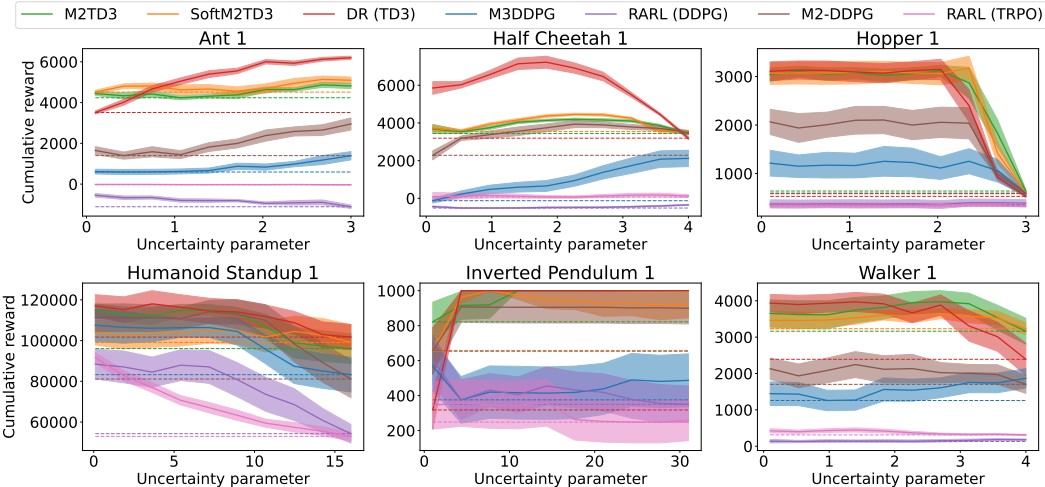

Figure 4: Cumulative rewards under trained policies for each uncertainty parameter $\omega \in \Omega$. The average (solid line) and standard error (band) for each $\omega \in \Omega$, as well as the worst average value (dashed line) are shown.

behavior policies $\beta$ and $\alpha$, respectively. Let $\rho_\alpha^\beta(s) = \lim_{T \to \infty} \frac{1}{T} \Sigma_{t=0}^{T-1} \int_{s_0} q_{\alpha,\beta}^t(s \mid s_0) p_0(s_0) \mathrm{d}s_0$ represent the stationary distribution of $s$ under $\beta$ and $\alpha$, where the step-$t$ transition probability density $q_{\alpha,\beta}^t$ is defined as $q_{\alpha,\beta}^t(s'|s) = \int_{a^p \in A_p} \int_{a^a \in A_a} q(s'|s, a^p, a^a) \beta(a^p|s) \alpha(a^a|s) \mathrm{d}a_a \mathrm{d}a_p$ and $q_{\alpha,\beta}^t(s'|s) = \int_{\bar{s} \in S} q_{\alpha,\beta}^{t-1}(\bar{s}|s) q_{\alpha,\beta}^1(s'|\bar{s}) \mathrm{d}\bar{s}$. The joint stationary distribution of $(s, a^p, a^a)$ is defined as $\rho_\alpha^\beta(s, a^p, a^a) = \beta(a^p|s) \alpha(a^a|s) \rho_\alpha^\beta(s)$.

We extend the action value function as a function of state $s$, protagonist agent's action $a^p$, and adversarial agent's action $a^a$, namely,

$$
\begin{aligned}
Q_\xi^\mu&(s, a^p, a^a) \\
&= \mathbb{E}[R_t \mid s_t = s, a_t^p = a^p, a_t^a = a^a, s_{t+k+1} \sim q(\cdot \mid s_{t+k}, \mu(s_{t+k}), \xi(s_{t+k})) \, \forall k \geqslant 0] \, , \quad (11)
\end{aligned}
$$

where $\mu$ and $\xi$ represent the policies of the protagonist agent and the adversarial agent, respectively, to be trained. We suppose that they are parameterized by $\theta_p$ and $\theta_a$, respectively.

The objective of M2TD3 is extended as

$$
\max_{\theta_p \in \Theta} \min_{\theta_a \in \Theta} J(\theta_p, \theta_a; \phi^*) \quad \text{s.t.} \quad \phi^* \in \operatorname*{argmin}_{\phi \in \Phi} L(\phi; \theta_p, \theta_a) \, , \quad (12)
$$

where the critic loss function $L$ is extended as

$$
L(\phi, \theta_p, \theta_a) := \int_{s \in S} \int_{a^p \in A_p} \int_{a^a \in A_a} (T_{\xi_{\theta_a}}^{\mu_{\theta_p}}[Q_\phi](s, a^p, a^a) - Q_\phi(s, a^p, a^a))^2
$$
$$
\times \rho_\alpha^\beta(s, a^p, a^a) \mathrm{d}s \mathrm{d}a^p \mathrm{d}a^a \, , \quad (13)
$$

where $T_{\xi_{\theta_a}}^{\mu_{\theta_p}}$ is a function satisfying

$$
T_{\xi_{\theta_a}}^{\mu_{\theta_p}}[Q](s, a^p, a^a) = r_\omega(s, a^p, a^a) + \gamma \int_{s' \in S} Q(s', \mu_{\theta_p}(s'), \xi_{\theta_a}(s')) q(s'|s, a^p, a^a) \mathrm{d}s' \, . \quad (14)
$$

The max-min objective function $J$ of the actor network is defined as

$$
J(\theta_p, \theta_a; \phi) := \int_{s \in S} Q_\phi(s, \mu_{\theta_p}(s), \xi_{\theta_a}(s)) \rho_\alpha^\beta(s) \mathrm{d}s \, , \quad (15)
$$

and $\rho_\alpha^\beta(s, a^p, a^a)$ and $\rho_\alpha^\beta(s)$ are approximated by the replay buffer $B$ that stores the trajectories obtained by the interaction using $\beta$ and $\alpha$. Let $\{(s_i, a_i^p, a_i^a, r_i, s_i')\}_{i=1}^M \subset B$ be mini-batch samples

taken uniformly randomly from the replay buffer. The approximated objective function used for the critic update is

$$\tilde{L}(\phi) = \frac{1}{M} \sum_{i=1}^{M} (y_i - Q_\phi(s_i, a_i^p, a_i^a))^2, \tag{16}$$

where $y_i = r_i + \gamma \cdot Q_\phi(s_i', \mu_{\theta_p}(s_i'), \xi_{\theta_a}(s_i'))$. The approximated objective function used for the actor update is

$$\tilde{J}(\theta_p, \theta_a) = \frac{1}{M} \sum_{i=1}^{M} Q_\phi(s_i, \mu_{\theta_p}(s_i), \xi_{\theta_a}(s_i)) \ . \tag{17}$$

Replacing (6) and (7) with above defined functions, we obtain M2TD3 for this setting.

**Experiment**   We used the tasks provided in [31][3]. For each trained policy, the worst-case performance is estimated by fixing the trained policy of the protagonist agent and training the adversarial agent's policy to minimize the protagonist agent's performance. To evaluate the worst-case performance of each approach, we performed 5 independent training for $2 \times 10^6$ time steps for each approach. The trial is indexed as $n \in \{1, \ldots, 5\}$. Then, for each obtained protagonist policy, we trained the adversarial policy for $2 \times 10^6$ time steps. We performed the adversarial policy training three times, and they are indexed as $m \in \{1, 2, 3\}$. During the training of the adversarial policy, we recorded the performance of the protagonist agent under the adversarial policy at time step $et \in \{10^5, 2 \times 10^5, \ldots, 2 \times 10^6\}$ (every $10^5$ time steps), denoted as $R(\mu, \xi_{m,et})$. Then, the worst-case performance of a protagonist policy $\mu$ was estimated by $R_{\text{worst}}(\mu) = \min_{m=1,2,3} \min_{et \in \{10^5, 2 \times 10^5, \ldots, 2 \times 10^6\}} R(\mu, \xi_{m,et})$. The average and standard error of $R_{\text{worst}}(\mu)$ over 5 trials are reported for each approach. The parameters and network architecture of the protagonist agent used in all methods and the adversarial agents used in M2TD3, M2-DDPG, M3DDPG, RARL (TD3), and RARL (TRPO) are the same as in the situation that the uncertainty parameter is directly encoded by $\omega$. In the situation that the uncertainty parameter was directly encoded by $\omega$, multiple uncertainty parameters were trained, but in this setting, only one adversarial agent was trained. The adversarial TD3 agents used to estimate worst-case performance were similar to the parameters and network architecture used in [7].

The result is shown in Table 9.

M2TD3 showed better worst-case performance than DR in all but the HalfCheetahAdv-v1 senarios. The reference performances were comparable to those of DR. Compared to those of RARL, M2TD3 showed competitive or superior performances, both in the worst-case and reference-case. Comparing M2-DDPG (the proposed approach based on DDPG instead of TD3) and M3DDPG, we observed similar performances in many scenarios in both reference-case and worst-case, and M2-DDPG significantly outperformed M3DDPG on HopperAdv-v1.

---

[3]https://github.com/lerrel/gym-adv

Table 9: Avg. $\pm$ std. error of reference performance (performance under no disturbance) and worst-case performance over five trials for each approach

| | TD3 | DR | M2TD3 | M2-DDPG | M3DDPG | RARL (TD3) | RARL (TRPO) |
|---|---|---|---|---|---|---|---|
| MuJoCo Environment: HalfCheetahAdv-v1 ($\times 10^4$) | | | | | | | |
| reference | $1.13 \pm 0.09$ | $1.21 \pm 0.03$ | $1.15 \pm 0.09$ | $1.19 \pm 0.04$ | $1.21 \pm 0.03$ | $1.04 \pm 0.09$ | $0.68 \pm 0.27$ |
| worst-case | $1.02 \pm 0.11$ | $1.12 \pm 0.05$ | $1.07 \pm 0.09$ | $1.13 \pm 0.06$ | $1.13 \pm 0.02$ | $1.00 \pm 0.10$ | $0.33 \pm 0.32$ |
| MuJoCo Environment: HopperAdv-v1 ($\times 10^3$) | | | | | | | |
| reference | $3.48 \pm 0.16$ | $3.55 \pm 0.14$ | $3.25 \pm 0.56$ | $2.63 \pm 0.52$ | $1.66 \pm 0.33$ | $2.42 \pm 0.94$ | $0.29 \pm 0.07$ |
| worst-case | $0.50 \pm 0.13$ | $1.79 \pm 1.25$ | $2.64 \pm 0.84$ | $2.16 \pm 0.53$ | $0.69 \pm 0.23$ | $0.82 \pm 0.45$ | $0.26 \pm 0.08$ |
| MuJoCo Environment: InvertedPendulumAdv-v1 ($\times 10^3$) | | | | | | | |
| reference | $1.00 \pm 0.00$ | $1.00 \pm 0.00$ | $1.00 \pm 0.00$ | $1.00 \pm 0.00$ | $1.00 \pm 0.00$ | $0.63 \pm 0.46$ | $0.10 \pm 0.07$ |
| worst-case | $0.03 \pm 0.01$ | $0.23 \pm 0.39$ | $0.83 \pm 0.35$ | $0.70 \pm 0.31$ | $0.80 \pm 0.38$ | $0.04 \pm 0.01$ | $0.03 \pm 0.01$ |
| MuJoCo Environment: SwimmerAdv-v1 ($\times 10^2$) | | | | | | | |
| reference | $1.42 \pm 0.09$ | $1.22 \pm 0.25$ | $1.38 \pm 0.09$ | $1.51 \pm 0.10$ | $1.51 \pm 0.06$ | $1.21 \pm 0.05$ | $0.21 \pm 0.10$ |
| worst-case | $0.96 \pm 0.15$ | $0.60 \pm 0.48$ | $1.14 \pm 0.08$ | $1.19 \pm 0.04$ | $1.21 \pm 0.04$ | $0.80 \pm 0.05$ | $-0.71 \pm 0.37$ |
| MuJoCo Environment: Walker2dAdv-v1 ($\times 10^3$) | | | | | | | |
| reference | $4.18 \pm 0.44$ | $4.27 \pm 0.61$ | $4.59 \pm 1.05$ | $2.81 \pm 0.81$ | $2.26 \pm 0.50$ | $3.95 \pm 0.27$ | $0.32 \pm 0.08$ |
| worst-case | $3.74 \pm 0.59$ | $3.61 \pm 0.93$ | $4.26 \pm 1.04$ | $1.75 \pm 0.62$ | $1.63 \pm 0.32$ | $3.11 \pm 0.13$ | $0.14 \pm 0.08$ |

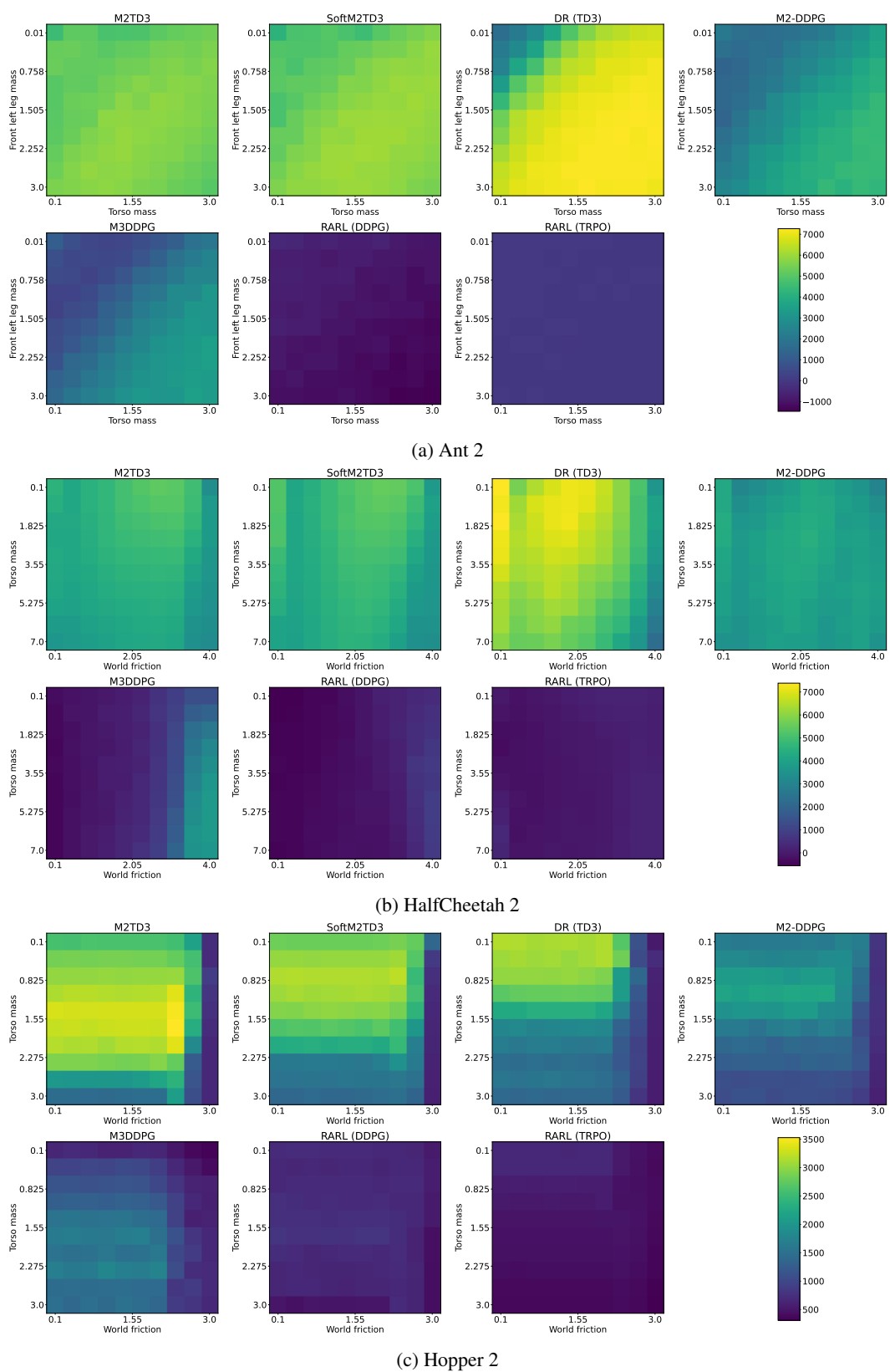

Figure 5: Cumulative rewards under trained policies for each uncertainty parameter $\omega \in \Omega$.

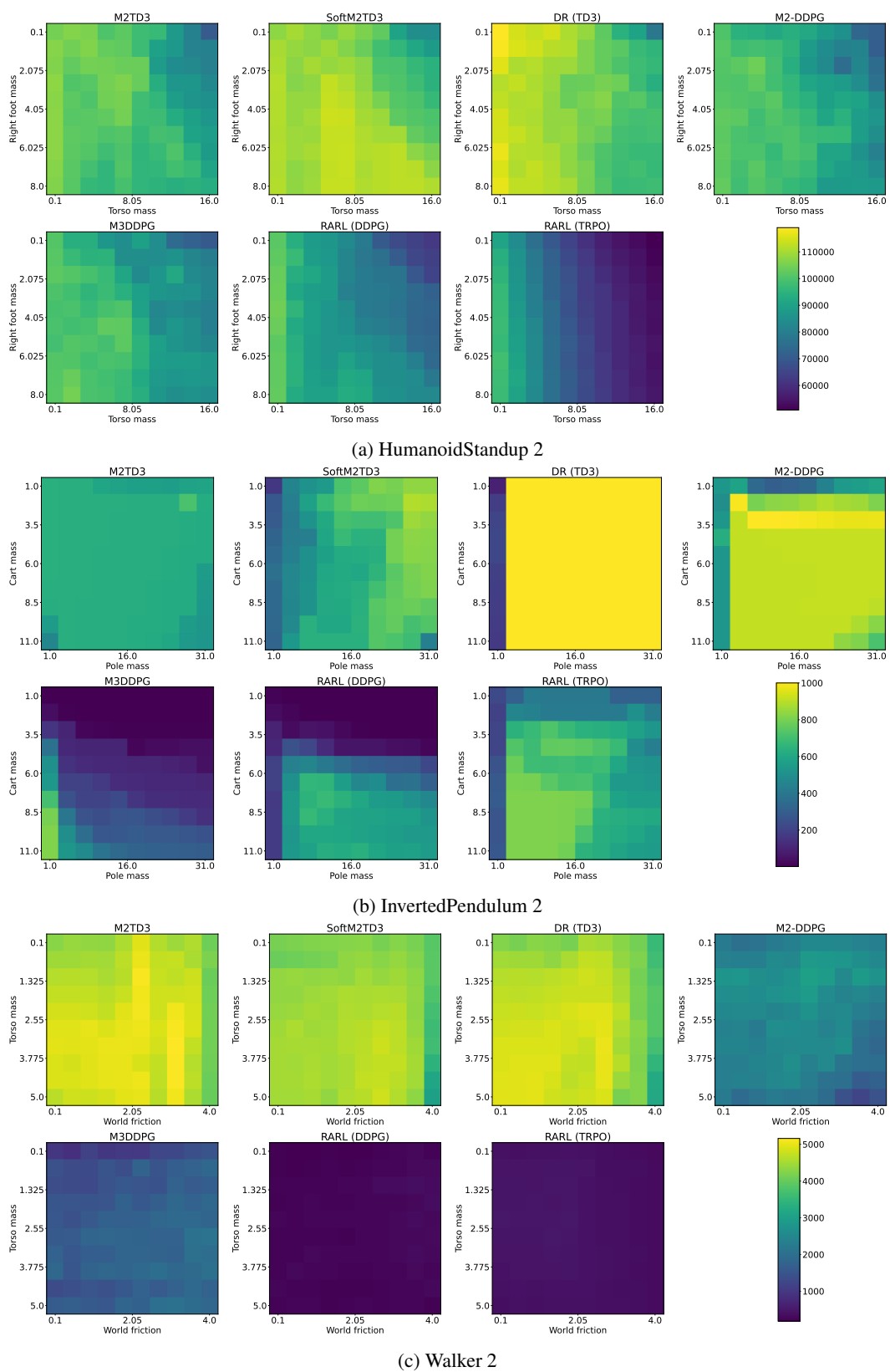

Figure 6: Cumulative rewards under trained policies for each uncertainty parameter $\omega \in \Omega$.

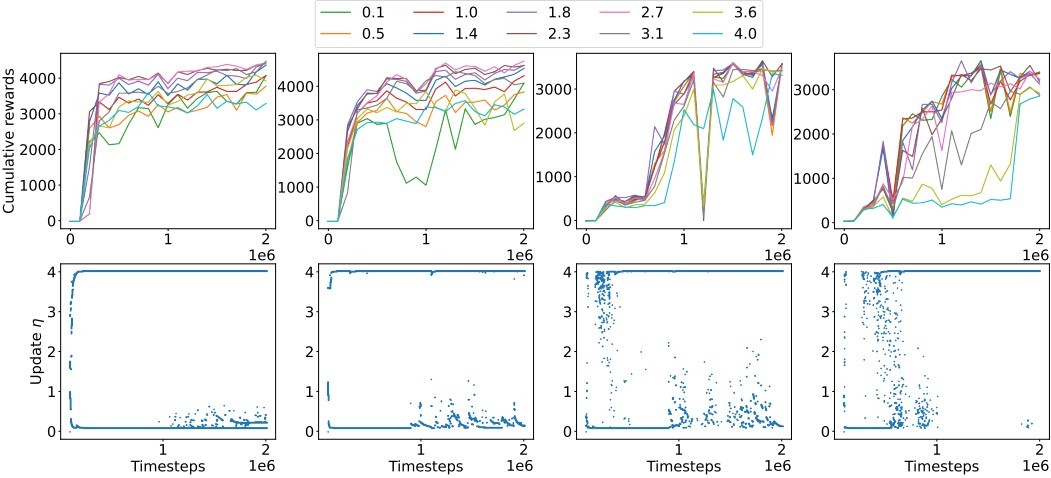

Figure 7: Cumulative reward under each uncertainty parameter (upper). Uncertainty parameter used to update at each time step (under). From left to right (scenario, algorithm): (HalfCheetah 1, M2TD3), (HalfCheetah 1, SoftM2TD3), (Walker 1, M2TD3), (Walker 1, SoftM2TD3).