# OpenReview forum: "Max-Min Off-Policy Actor-Critic Method Focusing on Worst-Case Robustness to Model Misspecification"
_NeurIPS.cc/2022/Conference — NeurIPS 2022 Accept_

### Official Review · Reviewer_tGzm · 2022-07-07

**Rating:** 5
**Confidence:** 4
**Soundness:** 2 fair
**Presentation:** 3 good
**Contribution:** 2 fair

**Summary:**

In this paper, the authors consider the gap from simulation environment to the real-world environment and aim to improve the robustness of policy. From a perspective of the worst-case performance on the uncertainty parameter set, they develop M2TD3 to solve a max-min optimization problem. In my opinion, this paper is well organized and the result is of abundant experiments. However, some problems should be concerned, mainly about the solution of the max-min optimization problem, the experiment settings and the selection of the baselines.

**Questions:**

Please see the questions in Weaknesses.

**Limitations:**

I think that this work would be more convincing if the authors can test the performance of M2TD3 with constantly changed uncertainty, because most of previous works are compatible with this setting. Further, what would be the impact on the algorithm if the fixed $\omega$ setting in one episode is removed.

**Strengths And Weaknesses:**

Strengths
	Clear problem formulation and very well related work classification;
	Abundant experiments and demonstratable results;
	Clear structure and understandable writing.

Weaknesses
	The authors claim that they develop a robust policy for sim2real scenario, but they suppose that the uncertainty parameter $\omega$ is fixed in each episode. I think this setting is more like transfer learning, which requires that the policy can adapt to tasks with different parameters. To make the policy more robust, the authors should consider the continuous changed uncertainty at each time step in one episode. And, in RARL, the uncertainty is present by an adversarial policy, which also interacts with the environment constantly. Thus, please explain the different setting compared with general robust RL.
	When solving the max-min optimization problem (3), the authors consider that the protagonist policy (robust policy) can observe the uncertainty (the same as the action of the adversarial policy defined in RARL) and add the uncertainty $\omega$ into the Q value function. In a sense, this observable uncertainty modelled in the Q value function makes the designed algorithm more like the domain randomization methods. And, the competitive results from DR(TD3) has the closest performance even DR(TD3) can exceed M2TD3 in the average performance. I think that the authors should add more detail comparison and analysis with DR(TD3).
	The proposed refresh strategy and the uncertainty parameter sampler may be some common methods for stable training policy and exploration noise for environment.
	Some new works should be considered.
[1] Jian P, Yang C, Guo D, et al. Adversarial Skill Learning for Robust Manipulation[C]//2021 IEEE International Conference on Robotics and Automation (ICRA). IEEE, 2021: 2555-2561.
[2] Yu L, Wang J, Zhang X. Robust Reinforcement Learning under model misspecification[J]. arXiv preprint arXiv:2103.15370, 2021.
[3] Zhang H, Chen H, Boning D, et al. Robust Reinforcement Learning on State Observations with Learned Optimal Adversary[C]//International Conference on Learning Representation (ICLR). 2021.
[4] Kamalaruban P, Huang Y T, Hsieh Y P, et al. Robust reinforcement learning via adversarial training with langevin dynamics[J]. Advances in Neural Information Processing Systems, 2020, 33: 8127-8138.
[5] Zhang H, Chen H, Xiao C, et al. Robust deep reinforcement learning against adversarial perturbations on state observations[J]. Advances in Neural Information Processing Systems, 2020, 33: 21024-21037.

---

> ### Author Response · Authors · 2022-08-02
> **Response to reviewer tGzm**
>
> Thank you very much for your valuable comment. There seems to exist reviewer's misunderstanding of the motivation and the setting of this paper. We hope the following replay clears up misunderstandings.
>
> > The authors claim that they develop a robust policy for sim2real scenario, but they suppose that the uncertainty parameter ω is fixed in each episode. I think this setting is more like transfer learning, which requires that the policy can adapt to tasks with different parameters. To make the policy more robust, the authors should consider the continuous changed uncertainty at each time step in one episode.
>
> > I think that this work would be more convincing if the authors can test the performance of M2TD3 with constantly changed uncertainty, because most of previous works are compatible with this setting. Further, what would be the impact on the algorithm if the fixed ω setting in one episode is removed.
>
> We do not think we should include the evaluation under a continuously changing environment, as it is out of the scope of this paper. To clear up any misunderstandings, let us first describe two situations we are considering. 1. We manufacture many manipulators of the same standard. Then, we want to install the trained policy to these manipulators. However, due to errors in the manufacturing process,  manipulators that differ slightly from the ideal one are produced. We do not want to retrain the policy (transfer learning) for each produced manipulator. To guarantee the worst case performance of these manipulators, it is reasonable to think that $\omega$ is fixed during each episode, but they are different for each manipulator in $\Omega$. 2. The environment changes due to aging. However, the change of the environment during each episode is negligible if each episode is not so long. These two situations are natural and they are modeled by our assumptions. The evaluation in our experiment is also consistent with these settings. Changing the environment during each episode is itself of interest, but we are sure that this setting is out of the scope of this paper. Exclusion of this setting should not be a limitation of this paper.
> The target of transfer learning is typically a specific (test) environment, not the worst case performance. Therefore, the motivation is different.
>
> > And, in RARL, the uncertainty is present by an adversarial policy, which also interacts with the environment constantly. Thus, please explain the different setting compared with general robust RL.
>
> As we mention in the related work section (second paragraph), the setting in RARL, where state-dependent external force is modeled by an adversarial policy and the adversarial policy is trained to make the main policy’s performance the worst, is covered by our setting. It is simply because we can consider the parameter of the adversarial policy as the uncertainty parameter $\omega$.
> In the revised manuscript, we add the experiment under the above setting in Appendix J. In this appendix, we also describe how the above setting is treated in the proposed approach in detail.
>
> > When solving the max-min optimization problem (3), the authors consider that the protagonist policy (robust policy) can observe the uncertainty (the same as the action of the adversarial policy defined in RARL) and add the uncertainty ω into the Q value function. In a sense, this observable uncertainty modeled in the Q value function makes the designed algorithm more like the domain randomization methods. And, the competitive results from DR(TD3) has the closest performance even DR(TD3) can exceed M2TD3 in the average performance. I think that the authors should add more detail comparison and analysis with DR(TD3).
>
> The obvious difference between DR and M2TD3 is that DR is maximizing the average performance while M2TD3 is maximizing the worst-case performance. The difference therefore clearly appears in Table 1 (worst-case performance) and Table 2 (average performance). Note that the table numbers are changed due to the revision. In addition, we compared the performance on each uncertainty parameter value (Figures 1, 4, 5, 6). Could you please provide a specific direction that the reviewer think is weak in terms of the comparison?
>
> > The proposed refresh strategy and the uncertainty parameter sampler may be some common methods for stable training policy and exploration noise for environment.
>
> We couldn’t find the reference that uses the refresh strategy for uncertainty parameters in the literature. Could you please provide the reference if this is really the weakness of the paper and should be addressed?
>
> > Some new works should be considered.
>
> Thank you very much for your suggestion. We add more recent paper in the related work.

---

> > ### Comment · Reviewer_tGzm · 2022-08-08
> > **Further discussion to the response.**
> >
> > Thanks for the author's reply. Some responses to my confusion are helpful. But, I still have some concerns about the settings and the proposed strategies.
> >
> > * The proposed scenarios are indeed consistent with the assumptions in this paper. But, I can not agree with the topic "Worst-Case Robustness to Model Misspecification" in the proposed situation. First, in some classic papers like RARL, M3DDPG, and Adv-SAC, the adversarial agent would generate action as a disturbance or a random parameter to the system and the protagonist agent needs to ensure the performance simultaneously. Thus, the settings in this paper may be a potential form of degradation compared with the previous settings, as a fixed-constant adversarial policy in each episode. Second, if the unknown parameters are fixed in each episode,  some methods like system identification, and disturbance observer are also useful in these settings so that we can estimate the unknown parameters and improve the control performance together. Based on the above concerns, I still suggest that the authors should discuss the continuously changed uncertainty case.
> >
> > * I am sorry for the unclear concern about the performance comparison between DR and M2TD3. I agree that it is because M2TD3 focuses on the worst-case performance guarantee, so M2TD3 present better performance than DR3 in the worst-case comparison. But, I notice that in Table 1, the performance of the robust RL algorithms (M2-DDPG, M3DDPG, and RARL) is far below DR. Thus, the authors may need to explain these results.
> >
> > * For the proposed refreshing strategy and the uncertainty parameter sampler, the response in g5qE can help understand the effectiveness of the two mechanisms. But, I suggest that the authors may detail formulate the relationship with the  tri-level optimization problem (The current version seems a little intuitive)

---

> > > ### Author Response · Authors · 2022-08-09
> > > **Response to reviewer tGzm**
> > >
> > > Thank you very much for your valuable comments.
> > >
> > > > Thus, the settings in this paper may be a potential form of degradation compared with the previous settings, as a fixed-constant adversarial policy in each episode.
> > >
> > > In general, there is a trade-off between the range of robustness guaranteed and the utility.
> > > Fixed-constant adversarial policy in each episode deal with a smaller range of robustness than the settings addressed by classic papers such as RARL, M3DDPG, and Adv-SAC.
> > > Therefore, these settings are not simply a degraded form compared to previous settings.
> > >
> > > > Second, if the unknown parameters are fixed in each episode, some methods like system identification, and disturbance observer are also useful in these settings so that we can estimate the unknown parameters and improve the control performance together. Based on the above concerns,
> > >
> > > We agree that if the unknown parameters are fixed for each episode, methods such as system identification can be useful.
> > > However, system identification usually introduces estimation errors.
> > > Therefore, we think that the scenarios addressed in this study can be combined with these.
> > >
> > > > I still suggest that the authors should discuss the continuously changed uncertainty case.
> > >
> > > Indeed, We think it is important to additionally discuss the case of continuously changed uncertainty.
> > > Therefore, please see Appendix J of Supplementary Material, which has been added for such settings.
> > >
> > > > But, I notice that in Table 1, the performance of the robust RL algorithms (M2-DDPG, M3DDPG, and RARL) is far below DR. Thus, the authors may need to explain these results.
> > >
> > > The following reasons clearly explain why the worst-case performance of the robust RL algorithm is lower than that of DR.
> > >
> > > M2-DDPG : M2-DDPG is the DDPG version of the off-policy actor critic algorithm from TD3, which is based on the proposed M2TD3 method.TD3 solves several problems in DDPG and can perform better than DDPG.
> > > Since the base algorithm used for DR is TD3, the scenario in which DR outperforms M2-DDPG can be attributed to the performance difference between TD3 and DDPG.
> > >
> > > RARL and M3DDPG : RARL and M3DDPG are methods to obtain robust policy by maxmin optimization.
> > > However, these methods may overfit local worst-case scenarios, in which case the performance of the policy may be excessively degraded.
> > > Therefore, it is likely that RARL and M3DDPG had lower worst-case performance than DR in many scenarios.

---

> > > > ### Comment · Reviewer_tGzm · 2022-08-09
> > > > **Thanks for the response**
> > > >
> > > > Thanks for the authors' extra description and experiments in Appendix J. For the current version, I'm keeping the same score.

---

### Official Review · Reviewer_Lcvb · 2022-07-11

**Rating:** 7
**Confidence:** 4
**Soundness:** 3 good
**Presentation:** 3 good
**Contribution:** 3 good

**Summary:**

This work is concerned with robustness to model misspecification in reinforcement learning. The authors propose a modified objective function to the Twin-Delayed Deep Deterministic Policy Gradient (TD3) algorithm, whereby the critic takes as input an uncertainty parameter of the environment, and the actor optimizes the policy for a worst-case realization of the uncertainty parameters in a predetermined set. The latter optimization is done simultaneously in a gradient descent-ascent fashion as opposed to prior work (Robust Adversarial RL — RARL) which performs each optimization completely alternatingly. The work is validated by thorough experimental results that show improved worst-case performance of the proposed approach compared to a variety of relevant baselines.


**Questions:**

Some clarifying questions that I had for the authors:
- L68: what’s the difference between $q_\omega$ and $p_\omega$ defined on L60?
-  Could you comment on the differences between (5) and (2). I think there should be some form of correction factor in Eqn (5) as $\eta$ is different from $\omega$ sampled from $\alpha$ (similar to importance sampling in off-policy). I'm also wondering if using the state distribution according to randomly sampled $\omega$ would lead to instability because the underlying MDP is always changing, as compared to Eqn (2) where $\omega$ is fixed for the state data and the Q function.
- L229 - L235: Why aren’t all $\eta$’s updated at each step instead of only the worst one? Is it to keep diversity of $\eta$’s?



**Limitations:**

I think the authors do a reasonable job of discussing and addressing limitations of the work.
- One other limitation that I see is in designing of the set $\Omega$ such that $\omega^* \in \Omega$, and also to have $\omega$ be a configurable parameter in the simulation environment. This is common practice in robust approaches, however.

- Also as noted by the authors, small dimensions (1-3) of uncertainty parameters were tested due to the exponentially growing size of the uncertainty parameter set with increasing dimensions. Two improvements in this area could be to design the set of parameters in a more directed manner (than a uniform cover), or to do the search in this space could more efficiently than exhaustive search.

**Strengths And Weaknesses:**

Application of robust RL to the deep learning regime is an ongoing open problem that has been addressed to some extent in this work. I find the approach reasonable, well-explained, and placed in the context of the existing literature. The empirical analysis is also quite thorough. On the other hand, I feel a weakness of the work is in its similarity to RARL. Although the proposed approach of simultaneous gradient ascent-descent is reasonable, an advantage of this over RARL would be in better optimization. However this is not analyzed theoretically or experimentally. The authors do mention that the difficulty in optimization may explain why Domain Randomization (DR) demonstrates better worst-case performance than M2TD3, and I feel that this highlights a pretty big problem with the proposed approach: the optimization can become too difficult to achieve the stated objective. As a result, I feel that analyzing this aspect of the work would greatly strengthen the reasons for adoption of this algorithm.

Summary of points and some additional notes are provided below.

Strengths:
- Thorough empirical analysis
- Details of algorithm well-explained and sensible
- I felt that the authors explained the limitations and did sufficient comparison with existing literature

Weaknesses:
- Need to be able to set simulator settings randomly, and we get robustness only to these settings. This is common in the literature but it still faces similar issues as domain randomization, etc., in that simulation to real setting changes need to be known in advanced.
- There is no theoretical analysis of the convergence properties of the proposed algorithm for simple settings. These results may be impossible in the deep neural network regime but it would inspire confidence in the algorithm to have some result showing that we can expect convergence with direct or linear parameterization (similar to those done by [1], [2], etc)

Writing:
- Would probably be better to put Related Works after describing the algorithm because it was hard to understand all the details that were being compared without knowing the algorithm first.
- L228: $\max_\eta J_t$ should be $\min_\eta J_t$ ?
- Notationally I found the use of $\omega$ and $\eta$ as the uncertainty parameters confusing. I think it would make the reading easier if only one letter is used.

Minor:
- abstract has a few grammatical errors
- L37 — extra “has been reported”
- L265: appears to have a typo “optimize theta and omega optimizes (2) …”

[1] Agarwal, Alekh, et al. "Optimality and approximation with policy gradient methods in markov decision processes." Conference on Learning Theory. PMLR, 2020.
[2] Qiu, Shuang, et al. "On finite-time convergence of actor-critic algorithm." IEEE Journal on Selected Areas in Information Theory 2.2 (2021): 652-664.

---

> ### Author Response · Authors · 2022-08-02
> **Response to reviewer Lcvb (1/2)**
>
> Thank you very much for your valuable comments and positive feedback.
>
> > I feel a weakness of the work is in its similarity to RARL. Although the proposed approach of simultaneous gradient ascent-descent is reasonable, an advantage of this over RARL would be in better optimization. However this is not analyzed theoretically or experimentally.
>
> The problem of alternating approach in RARL is described in the last paragraph of the related work section. In the case of the alternating approach, it has been theoretically shown to diverge in concave-convex quadratic functions. In the case of gradient ascent descent, it has been theoretically shown to converge to the optimal solution for the worst-case function on strongly convex and Lipschitz smooth functions if the learning rate is sufficiently small [16]. However, it is difficult to theoretically analyze the case in which critic learning is included, and we recognize that this is a limitation to be addressed in the future work.
>
> > The authors do mention that the difficulty in optimization may explain why Domain Randomization (DR) demonstrates better worst-case performance than M2TD3, and I feel that this highlights a pretty big problem with the proposed approach: the optimization can become too difficult to achieve the stated objective. As a result, I feel that analyzing this aspect of the work would greatly strengthen the reasons for adoption of this algorithm.
>
> We totally agree with this comment. This is why we introduced a variant of M2TD3, namely SoftM2TD3, that makes the optimization problem similar to DR while managing the worst case performance. Unfortunately, we have not yet managed to analyze this perspective further. It would be nice if you could suggest any direction to investigate this perspective. Any comments are appreciated.
>
> > Need to be able to set simulator settings randomly, and we get robustness only to these settings. This is common in the literature but it still faces similar issues as domain randomization, etc., in that simulation to real setting changes need to be known in advanced.
>
> > One other limitation that I see is in designing of the set Ω such that ω∗∈Ω, and also to have ω be a configurable parameter in the simulation environment. This is common practice in robust approaches, however.
>
> We agree. This is, however, out of the scope of the current paper, and we will address it in the future work.
>
> > There is no theoretical analysis of the convergence properties of the proposed algorithm for simple settings. These results may be impossible in the deep neural network regime but it would inspire confidence in the algorithm to have some result showing that we can expect convergence with direct or linear parameterization (similar to those done by [1], [2], etc)
>
> Thank you very much for the suggestion. We will investigate this direction in the future work.
>
> > Would probably be better to put Related Works after describing the algorithm because it was hard to understand all the details that were being compared without knowing the algorithm first.
>
> Thank you for the suggestion. We follow this suggestion in the revised manuscript.
>
> > Notationally I found the use of ω and η as the uncertainty parameters confusing. I think it would make the reading easier if only one letter is used.
>
> We intentionally used different letters to avoid confusion. But following the suggestion, we change the notation from $\eta$ to $\hat{\omega}$.
>
> > L68: what’s the difference between $q_\omega$ and $p_\omega$ defined on L60?
>
> The transition probability is denoted by $p_\omega$, whereas $q_\omega$ is the probability of the next state that takes into account the re-initialization of the state due to the end of the episode. That is, $q_\omega(s_{t+1} \mid s_{t}, a_{t}) =  p_\omega(s_{t+1} \mid s_{t}, a_{t})$ if $h_t = 0$; otherwise $q_\omega(s_{t+1} \mid s_{t}, a_{t}) = p_0(s_{t+1})$.

---

> ### Author Response · Authors · 2022-08-02
> **Response to reviewer Lcvb (2/2)**
>
> > Could you comment on the differences between (5) and (2). I think there should be some form of correction factor in Eqn (5) as η is different from ω sampled from  α
>
> The difference is commented in the paragraph after Eq. (5). First, the ground truth Q-function is replaced with a critic. The latter is expected to approach the former by minimizing Eq. (4). Second, the stationary distribution is taken for $\omega \sim \alpha$ in Eq. (5), whereas it is taken for a specific $\omega$ in Eq. (2). Ultimately, we want to maximize Eq. (2). However, it is difficult to estimate Eq. (2) using the replay buffer. Therefore, we used Eq. (5). In the middle of optimization, there exists a discrepancy between Eq. (2) and Eq. (5). However, once $\alpha$ is concentrated around the estimates of the worst $\omega$, the replay buffer will be filled with samples drawn for these $\omega$ values, and Eq. (5) will coincide with Eq. (2). Therefore, we have not introduced a correction factor in Eq. (5). Instead, we make $\alpha$ concentrated around the estimated worst $\omega$ at the end of training, as described in the paragraph “Uncertainty Parameter Sampler” and in Appendix C.
>
> > I'm also wondering if using the state distribution according to randomly sampled  ω would lead to instability because the underlying MDP is always changing, as compared to Eqn (2) where ω is fixed for the state data and the Q function.
>
> Whether Eq. (5) or Eq. (2) is used, the environment changes always due to the worst $\omega$ search. Therefore, the instability should appear both for Eq. (5) and Eq. (2) until the worst $\omega$ is more or less determined and $\alpha$ is concentrated around them.
>
> > L229 - L235: Why aren’t all η’s updated at each step instead of only the worst one? Is it to keep diversity of η’s?
>
> The main reason is that typically only one $\eta$ (now denoted by $\hat{\omega}$) contribute to the worst case performance and its gradient is nonzero only for such $\eta$. Therefore,
> updating only one $\eta$ is arguably the most natural way from the gradient-based optimization viewpoint. We can still update all $\eta$ using the same gradient, but it will lead to convergence of multiple $\eta$ to a single point, as the reviewer probably expects.
>
> > Also as noted by the authors, small dimensions (1-3) of uncertainty parameters were tested due to the exponentially growing size of the uncertainty parameter set with increasing dimensions. Two improvements in this area could be to design the set of parameters in a more directed manner (than a uniform cover), or to do the search in this space could more efficiently than exhaustive search.
> Thank you for the suggestion. Indeed, we tried estimating the worst parameters by using CMA-ES, a black-box optimizer. However, possibly due to the noise and/or multimodality in the cumulative reward, the worst parameter estimation by CMA-ES was not successfully converged. Therefore, we used a simple grid search.
>
> Please also see Appendix J in the revised manuscript, where we add a new experiments under the situation that the uncertainty parameter models an adversarial policy of the external force. In this setting, we evaluate the worst case performance of each trained policy by training the adversarial policy to minimize the protagonist performance while fixing the protagonist policy. In this situation the number of parameters to be optimized for the worst-case performance estimation is much greater than 3 (as it is the number of parameters in the adversarial policy).
>
> > L228: $\max_\eta$ should be $\min_\eta$?
>
> > abstract has a few grammatical errors
>
> > L37 — extra “has been reported”
>
> > L265: appears to have a typo “optimize theta and omega optimizes (2) …”
>
> Thank you very much for the correction.

---

> > ### Comment · Reviewer_Lcvb · 2022-08-09
> > **Response to authors**
> >
> > I thank the authors for their detailed response. Most of my questions have been answered.
> >
> > >Please also see Appendix J in the revised manuscript, where we add a new experiments under the situation that the uncertainty parameter models an adversarial policy of the external force. In this setting, we evaluate the worst case performance of each trained policy by training the adversarial policy to minimize the protagonist performance while fixing the protagonist policy. In this situation the number of parameters to be optimized for the worst-case performance estimation is much greater than 3 (as it is the number of parameters in the adversarial policy).
> >
> > The setting of having an adversarial policy applying an external force is quite different from the more general model misspecification setting, where any part of the dynamics could be chosen adversarially. See "Tessler, Chen, Yonathan Efroni, and Shie Mannor. "Action robust reinforcement learning and applications in continuous control." International Conference on Machine Learning. PMLR, 2019".
> >
> > I also think many of the other reviewers’ objections (not all of them, but many) are due to misunderstandings of the setting considered in the paper. Having read all the discussions, I am increasing my score from 6 to 7.
> >
> > Regarding the discussion of the relative weakness of a time-varying uncertainty set vs. one that is fixed over episodes, perhaps the authors could use arguments similar to those in [Nilim, Arnab, and Laurent El Ghaoui. "Robust control of Markov decision processes with uncertain transition matrices." Operations Research 53.5 (2005): 780-798.] to argue that the settings are not so different when the horizon is long.

---

> ### Comment · Area_Chair_rojv · 2022-08-09
> **Thank you! Are you satisfied by the answers?**
>
> Dear reviewer,
>
> Thanks again for your detailed review! The authors have responded to the reviews. Have they answered your questions satisfactorily? If not, what are the remaining issues?
> If you have any further questions from them, please ask them now. The deadline for discussion between the reviewers and the authors is today (Tuesday, August 9th).
>
> Since you are one of the more positive reviewers on this paper, I'd like to know your reaction regarding other reviews.
>
> Thank you,
> Area Chair
> P.S: Also as a courtesy to the authors, please acknowledge their rebuttal.

---

### Official Review · Reviewer_g5qE · 2022-07-11

**Rating:** 5
**Confidence:** 3
**Soundness:** 3 good
**Presentation:** 2 fair
**Contribution:** 2 fair

**Summary:**

This work studies the problem of optimizing the worst-case performance in a given uncertainty parameter set.  They propose an off-policy DeepRL algorithm for this problem setting (M2TD3) that solves a tri-level optimization problem using GAN-like gradient descent approach. They show the effectiveness of their proposed approach on a variety of MuJoCo tasks against different baselines.

**Questions:**

- Can you please expand on the motivation on the formulation? Please check the questions in the weaknesses above.
- Is there any formal claim of statement in Line 182?
- The part about the uncertainty parameter sampler is not clear. Can you please expand on it more (lines 248-251)?

**Limitations:**

The authors should discuss more the implications of assumption in lines 82-84.

**Strengths And Weaknesses:**

# Stregths
- Strong empirical results: The proposed approach has strong empirical results, and the comparison with the baselines seems thorough.

# Weaknesses
- Formulation: Certain aspects of the formulation are not explained in detail. For instance, the choice of only considering deterministic policies when the behaviour policy can be stochastic is not clear. Additionally, the assumption in lines 82-84 should be introduced as a separate assumption and accompanied by proper motivation. In the current form, there is a disconnect between the motivation in the introduction (Real-world cases) and the formulation and the empirical studies.

- Presentation: Most of the major contributions of the work seem to be algorithmic. However, most of the algorithmic details have been pushed to the Appendix. I think the related work and some of the tables could have been pushed to the Appendix to introduce the algorithm in the main text. Additionally, another baseline (softM2TD3) seems to work really well on these tasks however not many details are provided about it in the main text.

- Related work: I think RobustMDPs should also be included in the related work.

- Hyper-parameter overhead and reproducibility: The paper introduces many algorithmic components and new hyper-parameters, but there is not much discussion of the overhead of introducing these hyper-parameters ($\lambda_{\theta, \eta}, d_{tre}$). There is also no accompanying code for the work.

---

> ### Author Response · Authors · 2022-08-02
> **Response to reviewer g5qE (1/2)**
>
> Thank you very much for your valuable comment and positive feedback. There seems to exist reviewer's minor misunderstanding of the motivation of this paper. We hope the following replay clears up misunderstandings.
>
> > Additionally, the assumption in lines 82-84 should be introduced as a separate assumption and accompanied by proper motivation. In the current form, there is a disconnect between the motivation in the introduction (Real-world cases) and the formulation and the empirical studies.
>
> > The authors should discuss more the implications of assumption in lines 82-84.
>
> A probable source of the confusion is “not fixed over episodes” in the last paragraph of the introduction. We meant by this that the uncertainty parameter is fixed during each episode but it may vary episode by episode. This is consistent with the formal assumption written in Lines 42–44 and Lines 82–84. Situations we consider by this assumption is that the environment changes over time possibly due to aging, but each episode is short enough so that we can assume that the environment does not change during each episode.
>
> > Certain aspects of the formulation are not explained in detail. For instance, the choice of only considering deterministic policies when the behaviour policy can be stochastic is not clear.
>
> > Can you please expand on the motivation on the formulation? Please check the questions in the weaknesses above.
>
> The core idea of the method is to extend the critic and to apply the simultaneous gradient descent method on the extended critic. Though we used a deterministic target policy as the approach is built on top of TD3 and DDPG, we do not think there is a potential barrier when applying the proposed approach to other off-policy actor-critic approaches such as SAC, which uses a stochastic target policy. We will investigate the applicability in the future work.
>
> > Presentation: Most of the major contributions of the work seem to be algorithmic. However, most of the algorithmic details have been pushed to the Appendix. I think the related work and some of the tables could have been pushed to the Appendix to introduce the algorithm in the main text. Additionally, another baseline (softM2TD3) seems to work really well on these tasks however not many details are provided about it in the main text.
>
> Thanks for the suggestion. Instead of pushing the algorithmic details to the main text, we point to the corresponding Appendix section for each algorithmic component in the main text.
>
> > Related work: I think RobustMDPs should also be included in the related work.
>
> Thanks for the suggestion. Although ROPI is considered as a robust MDP approaches and is included in the related work, further Inclusion robust MDPs approaches in the related work may improve reader’s understanding of the difference between our setting and that of robust MDPs.
>
> > Hyper-parameter overhead and reproducibility: The paper introduces many algorithmic components and new hyper-parameters, but there is not much discussion of the overhead of introducing these hyper-parameters (λθ,η,dtre). There is also no accompanying code for the work.
>
> The code link was provided in Appendix A in the original submission, as we mentioned in the checklist. To emphasize it, it appears in the abstract in the revised manuscript.
> Regarding the hyper-parameter values, $\lambda_\theta$ and $\lambda_\eta$, we consider that they are not as sensitive as the learning rate values in GANs. For the hyper-parameter values inherited from TD3, we set the same values as used in the original paper of TD3.
> Here we provide the results of some parameter survey on $\lambda_\eta$, $d_{thre}$, and $p_{thre}$.
> The following table shows the performance of the proposed approach on HalfCheetah1 with $\lambda_\eta \in \{3e-5, 3e-4, 3e-3\}$. The average and the standard error of the worst case performance are shown. The other settings are the same as the experiment in the paper.
>
> | Senario                   | 3e-5        | 3e-4        | 3e-3         |
> | :-----------------------: | :---------: | :---------: | :----------: |
> | HalfCheetah 1($\times 10^3$)       | 3.17 ± 0.10 | 3.11 ± 0.15 | 3.10 ± 0.11  |
>
> Similarly, we report the effect of $d_{thre}$ in the following table. Here, we set $d_{thre} \in \{0.01, 0.05, 0.1\}$ and fix $p_{thre} = 0$.
>
> | Senario                   | 0.01        | 0.05        | 0.1         |
> | :-----------------------: | :---------: | :---------: | :---------: |
> | HalfCheetah 1($\times 10^3$)       | 3.16 ± 0.10 | 3.10 ± 0.08 | 3.21 ± 0.08 |
>
> The following table shows the effect of $p_{thre}$ in the following table. Here, we set $p_{thre} \in \{0.01, 0.05, 0.1\}$ and fix $d_{thre} = 0$.
>
> | Senario                   | 0.01        | 0.05        | 0.1         |
> | :-----------------------: | :---------: | :---------: | :---------: |
> | HalfCheetah 1($\times 10^3$)       | 3.27 ± 0.09 | 3.22 ± 0.06 | 3.22 ± 0.06 |

---

> > ### Comment · Reviewer_g5qE · 2022-08-08
> > **Thank you for your response,**
> >
> > I would like to thank the authors for their response. I also apologize for missing the link to the code repo. After reading their response, I still think there is a gap between the motivation and the formulation, in particular why this formalism for the problem makes sense is still not clear. The implication of assumptions is not discussed, the presentation of the algorithmic contributions of the work has not changed, and it is still unclear if the techniques presented are applicable to the general class of stochastic policies. Considering this, my stance on the work has not changed and I'm keeping the same score.

---

> ### Author Response · Authors · 2022-08-02
> **Response to reviewer g5qE (2/2)**
>
> > Is there any formal claim of statement in Line 182?
>
> In the recursive definition $Q^{\mu_\theta}(s, a, \omega) = r_\omega(s, a) + \gamma \mathbb{E}[Q^{\mu_\theta}(s', \mu_\theta(s'), \omega) \mid s' \sim q_\omega(\cdot\mid s, a)]$, if we consider the pair of $s$ and $\omega$ as a state, then this results in the standard Bellman equation. Line 182 is supported by this argument.
>
> > The part about the uncertainty parameter sampler is not clear. Can you please expand on it more (lines 248-251)?
>
> The detail was given in Appendix C. To make it clearer, we point to the corresponding part of the appendix from the main part describing the uncertainty parameter sampler.

---

### Official Review · Reviewer_ViYo · 2022-07-12

**Rating:** 4
**Confidence:** 3
**Soundness:** 2 fair
**Presentation:** 2 fair
**Contribution:** 2 fair

**Summary:**

This paper studies off-policy reinforcement learning, where there is a simulation environment with uncertainty parameters and the set of their possible values. The authors provided a solution of off-policy Actor-Critic which optimizes the worst-case performance and evaluated the proposed approach empirically.

**Questions:**

It would be helpful if the author could address my two main concerns: 1) lack of discussion/comparison of existing offline RL works, especially for the ones that share the same idea; 2) the empirical evaluation doesn't seem to match the goal/motivation of this work well.

**Limitations:**

see "Strengths And Weaknesses"

**Strengths And Weaknesses:**

This paper has a good motivation, which aims to utilize the uncertainty estimation in order to guarantee the well-behaving of the learning policy from the simulated environment to the real environment. The proposed solution in this paper is to that is to optimize the worst-case performance over policies.


The idea of optimizing the worst-case performance is actually not a novel idea in offline RL. There exist several existing works [1,2] that have explicitly discussed the same idea. Therefore, I feel it is needed to include the discussion of offline RL in the related work section.

In addition, the empirical evaluation doesn't seem to be able to verify whether the proposed algorithm achieves the main goal of this work---migrating the learning policy from the simulated domain to the real environment. It seems the experiments only focus on the final performance of some fixed environment, rather than the migration from simulation to the real world.

References:

[1] Bellman-consistent Pessimism for Offline Reinforcement Learning

[2] Provable benefits of actor-critic methods for offline reinforcement learning

---

> ### Author Response · Authors · 2022-08-02
> **Response to reviewer ViYo**
>
> Thank you very much for your valuable comment. We think that the negative evaluation is attributed to the reviewer’s misunderstanding of our experimental setting as described below.
>
> > In addition, the empirical evaluation doesn't seem to be able to verify whether the proposed algorithm achieves the main goal of this work---migrating the learning policy from the simulated domain to the real environment. It seems the experiments only focus on the final performance of some fixed environment, rather than the migration from simulation to the real world.
> > the empirical evaluation doesn't seem to match the goal/motivation of this work well.
>
> We are afraid that our experiment setting is misunderstood.
> In our experiment, we evaluated the worst-case performance (Table 1), and additionally the average performance (Table 2), of each trained policy rather than the performance in a specific environment. Please refer to the second paragraph of Experiment Setting section. As noted in the introduction, the real-world environment is assumed to be within a predetermined set of uncertainties, so the evaluation in this experiment represents a lower bound on the performance expected in the real-world environment. Therefore, the evaluation in our experiment matches the objective of this paper. Although this assumption has its own weaknesses, as the reviewer Lcvb mentioned in the first of the weaknesses, the reality is that many related studies currently make such an assumption.
> It is certainly important to test in a real environment. However, the real-world environment is only an example of what actually happened, and considering the purpose of evaluating the worst-case scenario, performance in the real-world environment does not necessarily represent the worst-case scenario. Fitting to a specific real-world environment is what the transfer learning does.
>
> > lack of discussion/comparison of existing offline RL works, especially for the ones that share the same idea
>
> Thank you for the suggestion. We add relevant studies on offline RL to the related work section.

---

> ### Comment · Area_Chair_rojv · 2022-08-09
> **Thank you! Are you satisfied by the answers?**
>
> Dear reviewer,
>
> Thanks again for your detailed review! The authors have responded to the reviews. Have they answered your questions satisfactorily? If not, what are the remaining issues?
> If you have any further questions from them, please ask them now. The deadline for discussion between the reviewers and the authors is today (Tuesday, August 9th). Also as a courtesy to the authors, please acknowledge their rebuttal.
>
> Thank you,
> Area Chair

---

### Meta-Review · Area_Chair_rojv · 2022-08-29

**Recommendation:** Accept
**Confidence:** Certain

**Metareview:**

This work proposes an actor-critic algorithm that optimizes the worst-case performance over a set of models. A key step is in incorporating the uncertainty parameter in the action-value function, a critic that estimates such a function, and an actor that searches over both a policy as well as the uncertainty parameter leading to the worst performance within the uncertainty set.

The algorithm is built on top of TD3. The paper provides several heuristics to ensure a good performance. It does not have any theoretical guarantee, which is a bit of concern for a work on robustness, but have a relatively comprehensive empirical results, showing the effectiveness of the proposed algorithm and its variants.

We have three reviewers on the accept side and one on the reject side: ViYo is a borderline reject, g5qE and tGzm are borderline accept, and Lcvb is an accept. Based on their review, and my own reading of the paper, I believe this paper can be accepted.

I explain some of the concerns and provide some suggestions to the authors.

ViYo's two main concerns are:
1. Paper not discussing related ideas in the offline RL literature.
2. The empirical evaluation not supporting the main goal of the paper: "migrating the learning policy from the simulated domain to the real environment"

The authors revised the paper and added some citations to the offline RL literature, so point (1) is addressed.

Regarding point (2), I disagree with the reviewer. It is true that the authors don't provide any experiment with real physical systems when sim2real is actually needed. But they have reported the worst-case performance of the obtained policy in the environment, which is the second best thing in this context, and is reasonable to me.
Since the reviewer did not engage with the authors to express their reaction to the response, I interpret the response as satisfying.

Some other concerns are:

- This work considers a fixed uncertainty parameter within an episode, as opposed to some other work on robustness that allows the uncertainty to change at every time step (tGzm).

This is a design choice of this paper, which I believe is reasonable. The authors revised the paper and added a footnote on p2 to clarify this.

- "Most of the algorithmic details have been pushed to the Appendix" (g5qE).

I agree with this. The paper regularly refers to an appendix for more details. Sometimes the detail can be easily postponed to an appendix, but sometimes they were just interrupting the flow of the paper. I believe having an algorithm box in the main body of the paper itself would be very helpful to understand how the method works.

A few other comments based on my own reading of the paper:
- Eq (6) is not the empirical version of Eq. (4), which is the Bellman error.
- Does M3DDPG algorithm reduces to this algorithm if the opponent-agent's policy mu_{\hat{w}} is defined to be a constant \hat{w}, as opposed to a state-dependent function?

As a general suggestion, I believe the clarity of the paper has some room for improvement. There have been a few misunderstandings in the reviews, some of them resolved and some not, which points me to the possibility that the paper might not be as clear as it can be.

**Award:**

No

---

### Decision · Program_Chairs · 2022-09-14

Accept